# Condensation of preformed charge density waves in kagome metals

Changwon Park [1] & Young-Woo Son [1] ✉

Charge density wave (CDW) is a spontaneous spatial modulation of charges in solids whose general microscopic descriptions are yet to be completed. Kagome metals of $AV_3Sb_5$ ($A$ = K, Rb, Cs) provide a chance to realize CDW intertwined with dimensional effects as well as their special lattice. Here, based on a state-of-the-art molecular dynamics simulation, we propose that their phase transition to CDW is a condensation process of incoherently preformed charge orders. Owing to unavoidable degeneracy in stacking charge orders, phases of preformed orders on each layer are shown to fluctuate between a limited number of states with quite slower frequencies than typical phonon vibrations until reaching their freezing temperature. As the size of interfacial alkali atom increases, the fluctuations are shown to counterbalance the condensation of orderings, resulting in a maximized transition temperature for $RbV_3Sb_5$. Our results resolve controversial observations on their CDWs, highlighting a crucial role of their interlayer interactions.

The kagome lattice has been considered as a fertile ground to realize exotic quantum phases because of its high degree of geometric frustration and nontrivial band structures[1–3]. Particularly, when the number of electrons per site is 2/3 ± 1/6, the Fermi energy ($E_F$) is at the van Hove singularity (vHS) such that the density of states diverges to induce various broken-symmetry phases[1–3]. The recent discovery of kagome metals of $KV_3Sb_5$, $RbV_3Sb_5$ and $CsV_3Sb_5$ (abbreviated as $AV_3Sb_5$ where $A$ denotes alkali atoms of K, Rb and Cs)[4] bring a chance to realize those phases because their $E_F$'s are close to vHS[4,5]. Indeed, several studies so far have demonstrated noteworthy states such as charge density waves (CDWs)[4,6–9], superconductivity[10], loop currents states[11–13] and electronic nematicity[14], to name a few.

On the other hand, a recent progress in manipulating stacked two-dimensional (2D) crystals opens a new window to study phase transitions[15–17]. Owing to their anisotropic lattice structures and interactions, distinct electronic phases are realized in the same material by pressure[18], a number of layers[17,19] and dopings[20] etc. Unlike simple lattice structures of hexagonal or rectangular shapes in typical layered materials[16], newly discovered kagome metals of $AV_3Sb_5$ add another complexity of frustrating intralayer orders in studying phase transitions, thus providing a unique chance to study their interplay with extreme anisotropic interactions.

Owing to the vHS in kagome metals, CDWs occur with decreasing temperature ($T$) from normal metallic phase[21,22]. In spite of their well-established presence, a few enigmatic observations raise questions on their formation mechanism. One of anomalous behaviors is the absence of phonon softening in inelastic X-ray[8], neutron[21] and Raman scattering experiments[23] while these results are incompatible with unstable phonon modes at $M$- and $L$-point of the Brillouin zone (BZ) in recent ab initio calculations[5,24,25]. Moreover, from the fact that the formation energy of CDW ($E_{CDW}$)[5] monotonically decreases with the size of the alkali atoms, one could naively expect that the transition temperature for CDW ($T_{CDW}$) increases accordingly. However, as the size of alkali atom increases from K to Rb and to Cs, the observed $T_{CDW}$ first increases from 78 K to 102 K and then decreases to 94 K[2], respectively, contradicting the conventional behaviors[26,27].

In this work, we theoretically showed that kagome metals first develop lattice distortions within each layer at much higher temperature of $T^*$ than $T_{CDW}$. Then, their phases are not fixed simultaneously with their amplitudes but fluctuate, which is shown to be inevitable owing to multiple ways of stacking CDWs between adjacent layers with the same energetic cost. As $T$ decreases further, the phases of preformed orders eventually stop varying at $T_{CDW}$. The characteristic timescale for the fluctuation is at least $10^5$ slower than that a typical

[1]Korea Institute for Advanced Study, Seoul 02455, Korea. ✉e-mail: hand@kias.re.kr

vibrational frequency of vanadium lattice. For CsV$_3$Sb$_5$, it is shown that a phonon softening signal exists only around $L$-point in a narrow temperature range around $T^*$. Between $T^*$ and $T_{CDW}$, the thermodynamic behaviors of preformed CDWs can be described by 4-states Potts model which undergoes a first-order phase transition at $T_{CDW}$[28]. Our quantitative analysis demonstrates that the fluctuations increase as the size of alkali metal increases, thus competing with $E_{CDW}$ for preformed orders. So, resulting $T_{CDW}$ maximizes not with Cs but with Rb atom and agrees well with the experimental trend[2], highlighting roles of weak interlayer interactions in determining phase transitions of layered kagome metals.

Our computations for two markedly disparate orders are made possible by developing a new large-scale molecular dynamics (MD) simulations method. A new method is specifically designed to describe lattice distortions using the relative displacements between atoms as basic variables (See Methods). Polynomial interatomic potentials are constructed using the linear regression for fitting about 2,500 basis functions to our first-principles calculation results. So, our method is accurate enough to reproduce our ab initio results very well, thus enabling us to perform large scale accurate lattice dynamics in a very long timescale.

## Results

### 2D charge orders and their interlayer interactions

Figure 1a shows the crystal structure of $A$V$_3$Sb$_5$. Vanadium atoms form 2D kagome lattice at vertices of dashed lines in Fig. 1b, c. One thirds of Sb atoms occupy the centers of dashed hexagons and the rest two thirds are at the upper and lower plane of the kagome lattice, respectively. Alkali atoms play as spacer atoms between the layers and determine the interlayer interactions. As shown in Fig. 1c, the CDW phase has been known to form the inverse star-of-David (iSOD) structure where V atoms in 2 × 2 units are modulated into one hexagon and two triangles instead of the star-of-David structure constructed by inverting the atomic displacements of the iSOD[5,25]. Our ab initio calculations also prefer the former over the latter, agreeing with previous studies[24,29]. We note that, without interlayer interactions, the 2 × 2 CDW in each layer can take any of the four translationally equivalent structures or phases indicated by four rhombi in Fig. 1c. The interlayer interaction, however, lifts the degeneracy between those random stacking phases. In Fig. 1d, we display possible stacking orders where the different phases are denoted by different colors corresponding to those in Fig. 1c.

We find that the same phases for neighboring layers are hardly realizable owing to the large energetic cost while the different phases are allowed and their energies for couplings are equivalent (See Supplementary Information Section 1). This implies that second-nearest-neighboring interlayer interactions should play an important role for stacking order. Without it, long-range stacking orders are absent and random stacking of CDWs would be dominant as long as the neighboring layers avoid to be the same phase (See detailed discussion in Supplementary Section 2), which contradicts experiments[8,10,30]. Our first-principles calculation estimates that its magnitude is order of 1 meV for 2 × 2 units in Fig. 1c. Such a small interaction may be related with possible CDW stacking faults or variations in interlayer ordering periods[8,10,23,30,31]. Notwithstanding its small magnitude, as we will demonstrate hereafter, it is an important interaction in freezing fluctuating charge orders of kagome metals.

From our MD simulations, we find that a interlayer correlation at temperature far above $T_{CDW}$ is not negligible in spite of the small negative long-rage interaction. Specifically, for CsV$_3$Sb$_5$ at $T = 130$ K, the probability that the pairs of second-nearest-neighboring layers have the same phases in the thermal ensemble turns out to be 0.66, revealing its crucial role (without it, the probability should be 0.25). From these considerations, the interlayer interactions are shown to have ambivalent characteristics depending on the interaction ranges such that nearest-neighboring layer should avoid the same phase while second-nearest-neighboring layer favors the same one.

Aforementioned formation of intralayer CDWs are captured explicitly by collecting the MD trajectories of V atom density of $\rho_V(\mathbf{r};T)$ for a given $T$. In Fig. 2a, it is shown that $\rho_V$ of CsV$_3$Sb$_5$ at $T = 200$ K with a simulation time of 0.4 nanoseconds perfectly match with ideal kagome lattice points. In enlarged views around the lattice point in Fig. 2b, it starts to deviate from the lattice points at $T \simeq 160$ K. As the temperature decreases further, it continuously deviates from the lattice points and the deviation reaches ~ 0.1 Å at $T = 20$ K.

### Phonon instability

The structural instability is also reflected in the temperature-dependent phonon dispersions. At finite $T$, the phonon spectrum can be extracted from the density-density correlation function of $S_{\rho\rho}(\mathbf{k}, \omega; T)$ where $\mathbf{k} = (k_x, k_y, k_z)$ is a crystal momentum and $\omega$ is a frequency of phonon (See Methods for details). In Fig. 2c, we plotted $S_{\rho\rho}(\mathbf{k}, \omega; T)$ of CsV$_3$Sb$_5$ along the high symmetric lines on $(k_x, k_y)$ plane with $k_z = b_z/2$ where $b_z$ is the out-of-plane reciprocal lattice vector. At

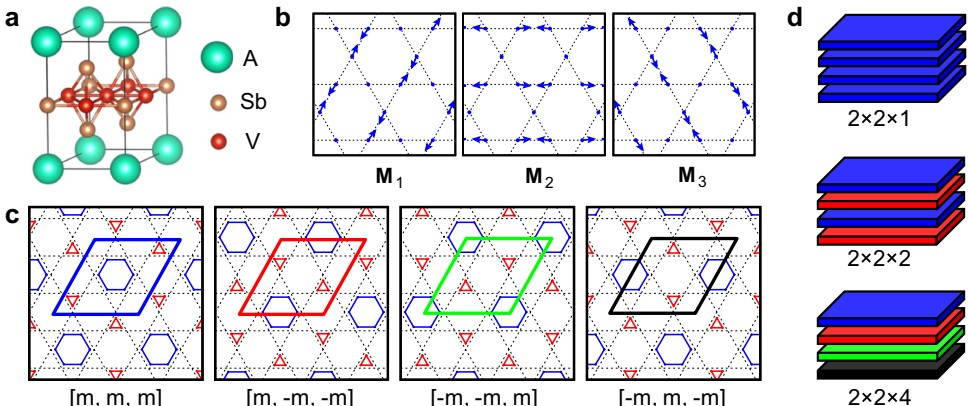

**Fig. 1 | 2 × 2 CDW of kagome metals and its low-energy stacking sequences.**
**a** Crystal structure of $A$V$_3$Sb$_5$ where $A$ denotes alkali atoms of K, Rb and Cs. **b** Three eigenmodes of the CDW at $M$-points of Brillouin zone (M$_1$, M$_2$, and M$_3$) for kagome shaped lattice of vanadium atoms. Blues dots are vanadium atoms and arrows denotes their displacements corresponding to each M$_i$($i$ = 1, 2, 3). **c** Schematic 2 × 2 CDW structures constructed from linear combinations of M$_i$'s where $[m_1, m_2, m_3] \equiv m_1$M$_1 + m_2$M$_2 + m_3$M$_3$. Kagome lattices (dashed lines) are modulated into hexagons (blue lines) and triangles (red) forming inverse star of David structure. There are four equivalent phases of 2 × 2 CDW (rhombi with four different colors). **d** Low-energy stacking sequences constructed by four phases in **c**. The phases of CDW are distinguished by colors matching those of rhombi in **c**.

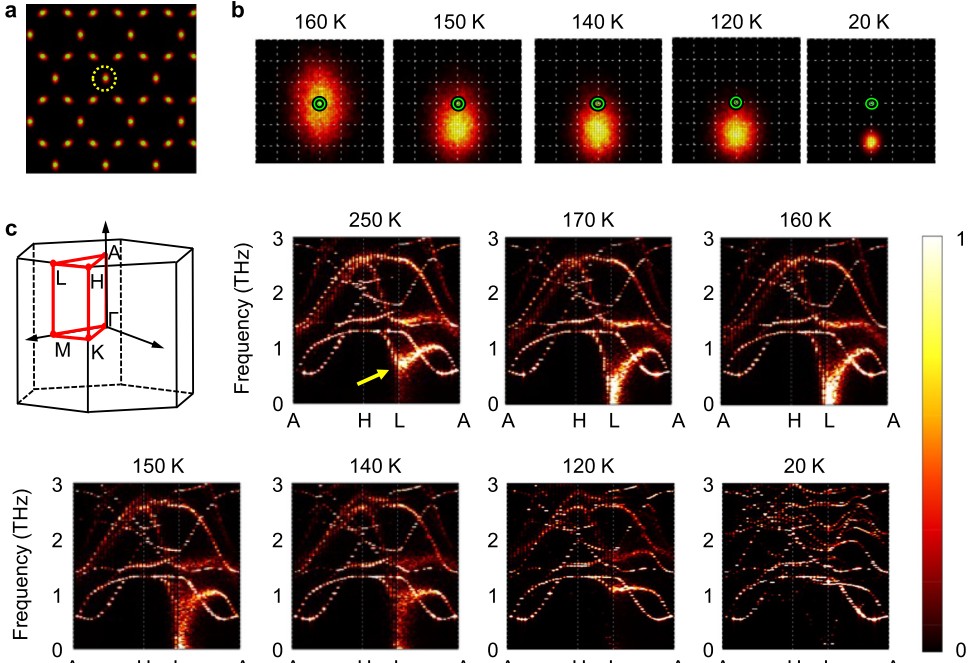

**Fig. 2 | Preformation of CDW in CsV$_3$Sb$_5$. a** Thermal distribution of vanadium atom density $\rho_V(\mathbf{r}; T)$ at $T = 200$ K. High (low) density regions are colored in yellow (black). **b** $\rho_V(\mathbf{r};T)$ within the dotted circle in **a** are enlarged at the temperature range from 20 to 160 K. Green open dots denote kagome lattice points and the space of the grid (dotted lines) is 0.05 Å. **c** Temperature-dependent phonon spectra obtained from $S_{\rho\rho}(\mathbf{k},\omega;T)$. In top left panel, Brillouin zone and high symmetric points of CsV$_3$Sb$_5$ are drawn. The spectrum at $T = 20$ K is multiplied by 10 for a visual clarity. The arrow in spectrum at $T = 250$ K indicates a soft mode. Acoustic phonon branches are invisible due to their weak intensities.

$T = 250$ K, a soft mode denoted by an arrow is clearly shown at $L$-point. As $T$ decreases, the frequency of the soft mode approaches zero. Eventually, at $T^* \simeq 160$ K, the structural instability develops, that may indicate a phase transitions with lattice distortions[32], compatible with our simulation of $\rho_V(\mathbf{r}; T)$ in Fig. 2b. However, the structural instability here does not guarantee phase transition. Even though the local $2 \times 2$ CDW orders within each layer and their $\pi$-phase shift between adjacent layers can fully develops, the local CDW orders change their phases dynamically and do not spontaneously break any symmetry of the crystal. As we will see later, a global broken symmetry state occurs at much lower temperature of $T_{CDW}$ so that we could interpret $T^*$ as the preformation temperature of CDW.

We note that right below $T^*$, all the softening signatures near $L$-point disappear as shown in Fig. 2c. At $T_{CDW} < T < T^*$, the preformed CDW domains have orders-of-magnitude longer lifetimes than vibrational periods as will be shown later. So, the effective phonon frequencies is essentially from the curvature of the potential around one of the four degenerate CDW phases. The phonon mode at $L$-point corresponding to this new metastable atomic configurations becomes harder as $T$ decreases such that near $T_{CDW}$, no softening signature can be found.

We also find that no soft modes exist along $\overline{\Gamma M}$ (Supplementary Fig. 2a), which is consistent with recent experiments reporting unexpected stability of phonon modes along $\overline{\Gamma M}$ at the transition[8,21,23]. Even at $T > T^*$, the lowest $M$-phonon remains hard unlike $L$-phonon, excluding a possible preformation of CDW. At this high temperature, anharmonicity-induced phonon hardening occurs for both phonons but the effect is much stronger for $M$-phonon due to the enhanced interlayer force constants. If not considering the anharmonic effect, both $M$- and $L$-phonon are expected to be unstable for the structure as were demonstrated by recent first principles calculations[5,24,25]. We can expect that the hardening effect for phonon at $U$-point of BZ ($2 \times 2 \times 4$ CDW) is in-between $M$- and $L$-phonon. Indeed, the softening of dispersions at $U$-point around $T^*$ is less conspicuous than one at $L$-point (Supplementary Fig. 2b), not fully developing into instability.

## Fluctuation of preformed orders

From the results so far, it seems that CsV$_3$Sb$_5$ undergoes a typical phase transition at $T^*$. However, owing to large degeneracy for pairing charge orders between layers, a slow fluctuation between them proliferates across the whole layers. We note that our simulation time of 0.4 nanoseconds for the apparent ordering is long enough for simulating usual structural transitions[33,34]. For kagome metals, it turns out to be not. With orders-of-magnitude longer simulations time of 12 nanoseconds, we indeed found a clear signature for the fluctuation. We also confirm that the fluctuation is not an artifact from the size effect of simulation supercell (See detailed discussions in Supplementary Section 4).

The phase fluctuations are unambiguously identified by eigenmode decomposition analysis. As shown in Fig. 1b, the in-plane CDW decomposes into three symmetrically equivalent eigenmodes of $\mathbf{M}_i(i = 1, 2, 3)$. It can be easily checked that their linear combinations of $m_1(t)\mathbf{M}_1 + m_2(t)\mathbf{M}_2 + m_3(t)\mathbf{M}_3$ can reproduce the four phases of $2 \times 2$ iSOD structures as shown in Fig. 1c. Here $m_i(t)(i = 1, 2, 3)$ is a time-dependent coefficient for eigenmodes of $\mathbf{M}_i$ where $t$ denotes time. By inspecting temperature-dependent time evolution of $m_i(t)$ for a specific $2 \times 2$ supercell, we can identify slow CDW fluctuations in kagome metals (see further details in Supplementary Section 5).

Figure 3a shows a temporal evolution of $m_1(t)$ for $T = 140$ K. In addition to picoseconds phonon vibrations, there are clear sign changes of $m_1(t)$ in timescale of a few nanoseconds without altering its averaged absolute magnitude. This indeed indicates phase flips of CDWs. The rate of phase flips can be quantified by computing a Pearson correlation coefficient (PCC)[35] for time interval of $\Delta t$, $r_{11}(\Delta t) \equiv \frac{E[m_1(t+\Delta t)m_1(t)] - E[m_1(t+\Delta t)]E[m_1(t)]}{\sigma[m_1(t+\Delta t)]\sigma[m_1(t)]}$, where $E[m_1]$ and $\sigma[m_1]$ are time average and standard deviation of $m_1$, respectively. As shown in Fig. 3b, the fast phonon vibrations and the slow phase flip are manifested as picoseconds oscillation and the subsequent slow decay, respectively (For comprehensive analysis of PCCs, see Supplementary Section 5). The decay rates of $r_{11}$ in Fig. 3b imply that the phase-coherence time does

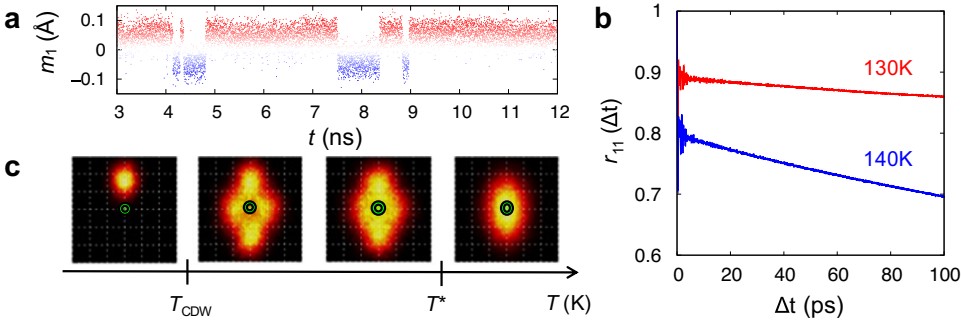

**Fig. 3 | Fluctuation of preformed CDW. a** Temporal fluctuation of the amplitude $m_1$ at a specific $2 \times 2$ supercell for $T = 140$ K. The sign changes correspond to phase flips of preformed CDW. **b** Temporal Pearson correlation coefficient of $r_{11}(\Delta t)$ at $T = 130$ and $140$ K. **c** Thermal evolution of $\rho_V(\mathbf{r}; T)$ incorporating slow phase flips. As the temperature decreases, the shape of $\rho_V(\mathbf{r}; T)$ changes from a Gaussian thermal ellipsoid (right panel, $T > T^*$) to four-peaked distribution (two middles, $T_{CDW} < T < T^*$), and finally be collapsed into one of the four peaks (left) at $T < T_{CDW}$.

not exceed a few nanoseconds at 140 K and becomes longer as $T$ decreases further. With inclusion of the phase flips, for $T_{CDW} < T < T^*$, the averaged local density profile of vanadium atoms of $\rho_V$ should change from a Gaussian thermal ellipsoid shown in Fig. 2b into a four-peaked or non-gaussian distribution as shown in Fig. 3c. As $T$ approaches $T_{CDW}$, each peak becomes to be more distanced and eventually, the fluctuations are quenched to one of the four peaks at $T = T_{CDW}$. This implies that in experiments incapable of resolving nanosecond dynamics, only the averaged properties will be observed without any symmetry-breaking feature between $T_{CDW}$ and $T^*$. However, those phase fluctuations are reflected as zero frequency peak at $S_{\rho\rho}(\mathbf{k}, \omega; T)$ (see Supplementary Fig. 3), that may be accessible by inelastic neutron scattering.

## Critical temperatures of charge orders

Our simulations hitherto reveal that in kagome metals, $T_{CDW}$ is nothing but a critical temperature for a globally ordered phase by condensing the CDWs that are preformed at $T^*(> T_{CDW})$. Although the physical process of the phase transition is now understood, a reliable quantitative MD calculation of $T_{CDW}$ is undoable because the time for quenching preformed orders well exceeds our attainable simulation time. This motivates us to map our systems into an anisotropic 4-states Potts model[28] to compute $T_{CDW}$ quantitatively using a well-established statistical method[36]. Due to the two well-separated time scales of dynamics (thermal vibration and phase fluctuation in Fig. 3a), if we average the atomic trajectories of preformed CDW's over 100ps, the averaged snapshot will look like one of four $2 \times 2$ CDW phase in Fig. 1c, but with temperature-dependent amplitudes. The four phases become 4-states 'spin' variables on lattice points of layered triangular lattices as shown in Fig. 4a. So, an effective Hamiltonian for the interacting spins therein can be written as

$$H = \sum_{\langle i,j \rangle, \alpha} J_{\parallel} \delta(s_{i,\alpha}, s_{j,\alpha}) + \sum_{i,\alpha} J_{\perp} \delta(s_{i,\alpha}, s_{i,\alpha+1}) + \sum_{i,\alpha} J_{\perp 2} \delta(s_{i,\alpha}, s_{i,\alpha+2}), \quad (1)$$

where $s_{i,\alpha}$ is an effective four-states spin at $i$-th site of $\alpha$-th layer, $J_{\parallel}$ their effective intralayer nearest-neighboring (n.n.) interaction, $J_{\perp}$ interlayer n.n. and $J_{\perp 2}$ every other interlayer n.n. interactions, respectively as described in Fig. 4a. Here, $\delta(s_{i,\alpha}, s_{j,\beta})$ is zero if states of two effective spins of $s_{i,\alpha}$ and $s_{j,\beta}$ are different, otherwise, it is one. We note from considerations above that signs and magnitudes of the interactions should be $J_{\parallel} \ll J_{\perp 2} < 0 < J_{\perp}$.

Unlike typical Potts models[28], interaction parameters in Eq. (1) are explicit functions of temperature because amplitudes of CDW vary as the temperature changes. For $T = 0$, $J_{\parallel}(0)$ and $J_{\perp}(0)$ can be readily calculated from the energy cost forming domain wall and energy

differences between different stacking, respectively (See detailed procedures in Supplementary Section 6). For $T > 0$, they are similarly obtained from thermally averaged potential energy of domain structure. Here, the thermal average is approximately calculated using harmonic and rigid approximations for density correlation functions (See Methods). Then, we have temperature-dependent exchange parameters of $J_{\parallel}(T), J_{\perp}(T), J_{\perp 2}(T)$ in our 4-states Potts model. So, to compute $T_{CDW}$, we need to solve the model self-consistently because temperature and the exchange parameters depend on each other. For kagome metals with alkali atoms of K and Rb, we can bypass time-consuming reference calculations by rescaling polynomial interatomic potential obtained for $CsV_3Sb_5$ without degrading accuracy (See Supplementary Section 7). We also note that the accuracy of $J_{\perp 2}$ from our ab initio calculation of $CsV_3Sb_5$ seems to be marginal. While the absolute magnitude of $J_{\perp 2}$ determines the ground state stacking structure, it turns out to play a minor role in determining $T_{CDW}$. So, we set it as a parameter of $J_{\perp 2} = -0.5J_{\perp}$ and checked that $T_{CDW}$ is lowered by at most 7 K when we decrease $J_{\perp 2}$ to be 0.

In Fig. 4b, c, the self-consistently obtained $J_{\parallel}(T)$ and $J_{\perp}(T)$ are shown for three kagome metals. The critical temperature for preformed orders of $T^*$ can be determined from a condition that the amplitude of intralayer CDW vanishes or that $J_{\parallel}(T)$ and $J_{\perp}(T)$ become zero simultaneously. As expected, the intralayer coupling of $J_{\parallel}(T)$ is largest (smallest) for $CsV_3Sb_5$ ($KV_3Sb_5$), being consistent with the lowest (largest) $E_{CDW}$[5]. Accordingly, $T^*$ monotonically increases as the size of alkali atom increases as shown in Fig. 4d.

The $J_{\perp}$ displays different temperature dependence compared with $J_{\parallel}(T)$. Since all the compounds here share the same kagome lattice layer formed by V and Sb atoms and differ by their alkali atoms, $J_{\perp}$ will be dominantly set by size of the alkali atoms. Indeed, the interlayer distance for Cs compound is longest among them (Supplementary Section 1) so that it has the smallest $J_{\perp}(0)$ as shown in Fig. 4c. This implies that the weakest $J_{\perp 2}$ of Cs compound will invoke the most severe fluctuation of CDW phase among three kagome metals. We also note that as temperature approaches $T^*$, the magnitude of $J_{\parallel}$ is comparable to those of $J_{\perp}$ as well as $J_{\perp 2}$. Therefore, we identify two competing features to determine thermodynamic states, i.e., the phase fluctuation of CDW counterbalances the $E_{CDW}$. So, even though $CsV_3Sb_5$ has the highest $T^*$, its $T_{CDW}$ can be lower than the other. Estimated $T^*$ and $T_{CDW}$ from Eq. (1) indeed confirm such a competitive interplay between inter- and intralayer orders, agreeing well with experimental trend as shown in Fig. 4d.

## Discussion

The preformed charge order discussed so far is not measurable as discontinuities in thermodynamics variables such as a specific heat or susceptibility. Also, we expect that Raman scattering[23,24,37,38] or nuclear

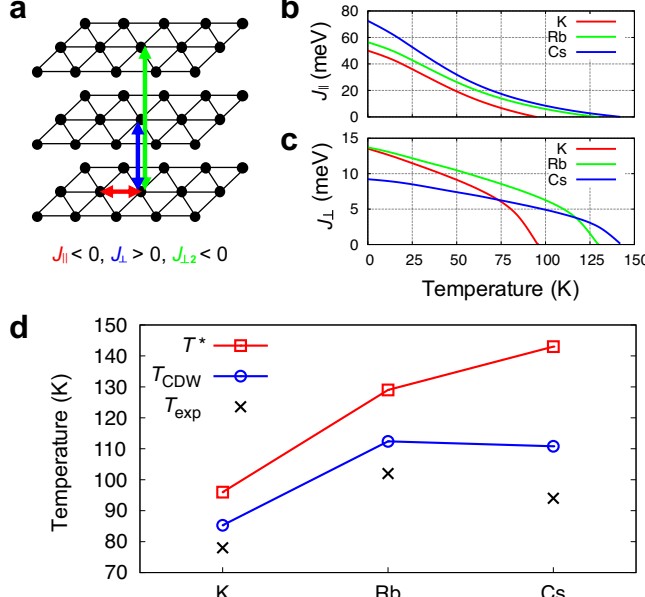

**Fig. 4 | Critical temperatures for CDW. a** Effective lattice structure for 4-state Potts model. Effective 4-state spins locate at vertices of triangles (black circles) corresponding to 2 × 2 supercells shown in Fig. 1**c**. Effective intralayer (red arrow), first- (blue arrow) and second-nearest-neighboring interlayer (green arrow) interactions between spins are denoted with $J_\parallel$, $J_\perp$ and $J_{\perp 2}$, respectively. **b, c** Temperature dependent $J_\parallel(T)$ and $J_\perp(T)$ of $A$V$_3$Sb$_5$ ($A$ = K, Rb, Cs), respectively. **d** Comparison of the experimental CDW temperatures ($T_{exp}$)[2] and calculated critical temperatures of $T^*$ (open squares) and $T_{CDW}$ (open circles) for different alkali atoms of $A$ on abscissa.

magnetic resonance measurements[39,40] may not be so easy to capture its explicit signals owing to its fluctuating phases and dynamical nature. But there might be at least three experimental signatures already indicating its existence. First, a recent X-ray diffraction experiment[22] reports that the integrated peak intensity of 2 × 2 × 2 CDW order in CsV$_3$Sb$_5$ survives up to 160 K, well above $T_{CDW}$ = 94 K. In addition to that, the thermal expansion of the in-plane lattice constant shows a slope change twice at $T_{CDW}$ and at $T$ = 160 K, respectively[22], implying the qualitative structural changes at higher $T$ than $T_{CDW}$. Second, the coexistence of the ordered and disordered phases near $T_{CDW}$ is measured through nuclear magnetic resonance measurements[39,40]. This is consistent with a first order phase transition from our anisotropic 4-states Potts model in Eq. (1). Third one is the absence of phonon softening around $T_{CDW}$[8,21,23]. As we already discussed, this absence is not anomaly but a consequence of preformation of CDW at much higher $T^*$.

In addition to these indirect evidences, we may have a way to detect fluctuating charge orders directly. Because the preformed CDW not only changes the vibrational properties but also introduces slow cluster dynamics, we expect that it can be also detected as a central peak in the dynamic form factor of density fluctuation, or apparent symmetry-forbidden signals in Raman scattering reminiscing the precursor formation in ferroelectric materials[41,42].

We note that our current MD methods from first-principles approaches do not show any state related with broken time-reversal-symmetry states. Within our methods, the vanadium $d$-orbitals have considerable out-of-plane band dispersion across the Fermi energy[43] so that spin-spilt states related with them hardly develop. From these, our MD could not touch upon the various experimental signatures[1–3,11–13] on time-reversal-symmetry broken phases. However, if exotic electronic and magnetic orders can couple to bond orders below $T_{CDW}$, we expect that our method could reflect the corresponding structural evolutions at lower temperature.

Lastly, we point out that the tiny magnitude of $J_{\perp 2}$ is essential in controlling the phase fluctuation with fixed CDW amplitudes across the kagome layers. This strongly suggests that a facile engineering of thermodynamic states of kagome metals could be possible through external controls of interlayer interactions[44,45] and that our current understanding of asynchronism in thermodynamics transitions of kagome metals can be applied to understand various subtle phase transitions in layered 2D crystals[15–20,46].

## Methods

### Construction of interatomic potential
Our interatomic potential is based on the linear regression with a set of polynomial basis functions which is adequate for small displacement phonon calculations[47]. We note that a similar method has been used for dynamical properties of kagome metal recently[48]. For the interatomic potential to meet a few physical conditions such as (1) continuous translational symmetry, (2) point and space group symmetry of the crystal, (3) permutation of variables, and (4) asymptotic stability for a large displacement, basis functions are constructed as follows.

First, to meet the condition (1), we replaced the variable from the atomic displacement $u_{\alpha_i}$ to the relative displacement $u_{\alpha_i} - u_{\alpha_i'}$ where $\alpha_i$ is a condensed index for Cartesian components, basis atom in a unitcell and the Bravais lattice point. The index of $\alpha_i'$ shares the same Cartesian component with $\alpha_i$ but has different basis atom index and Bravais lattice point, respectively. Then, the interatomic potential of $V$ can be written as

$$V = \sum_{n=0}^{\infty} V_n = \sum_{n=0}^{\infty} \frac{1}{n!} \sum_{\{\alpha_i, \alpha_i'\}} c_{\alpha_1 \alpha_1' \cdots \alpha_n \alpha_n'} \prod_{i=1}^{n} (u_{\alpha_i} - u_{\alpha_i'}), \quad (2)$$

where the brace indicates that the summation runs for all indices of $\alpha_i$ and $\alpha_i'$ in the summands. The condition (2) can be straightforwardly incorporated by applying all symmetry operations $R$ of a given crystal to $V_n$ in Eq. (2), that can be written as

$$\tilde{V}_n = \frac{1}{N_R} \sum_R R V_n,$$

where $N_R$ is a number of symmetry operations. The condition (3) will be automatically met from linear dependence check of basis function. The condition (4) is usually ignored in phonon calculations but becomes to be critical for potentials with a huge number of variables. Since our interatomic potential has thousands of variables, without explicit consideration of the condition (4), optimization processes do not end but diverge. In general, it is hard to prove that some multi-variable polynomial is bounded below but we avoid the negative divergence by forcing the highest-order basis function to be multiples of squares and by constraining their coefficients to be nonnegative during the linear regression process. Rigorously, though this approach does not guarantee the condition (4), we find that the $V$ in Eq. (2) constructed in this way did not cause any practical problem. The resulting basis functions are sorted in ascending order by the largest distance between the two atoms in the basis function and linearly-dependent basis functions are removed by Gram-Schmidt orthogonalization.

For CsV$_3$Sb$_5$, we truncated the polynomial out at the 4th order and cutoff the relative distance at 10 Å for the 2nd-order basis functions and 6 Å for 3rd- and 4th-order ones. Importantly, only V displacements are used for the variables in the 3rd- and 4th-order basis function because we found that the anharmonic effect is prominent only between the V atoms due to their large displacements in CDW phases. The number of basis functions made in this way is 434, 521, and 1453 for 2nd-, 3rd- and 4th-order, respectively.

The reference configurations are collected from multiple sources to ensure reliable sampling of potential energy surfaces. We

performed first-principles MD simulations of $4 \times 4 \times 2$ supercell at 100 and 200 K for 1600 steps with a time step of 10 fs. We then collected 53 configurations from each temperature that were equally spaced in time of 300 fs. Additional 30 configurations are collected from randomly displacing ($< 0.04$ Å) atomic positions from the optimized $2 \times 2 \times 2$ CDW structure. Lastly, the linearly interpolated configurations between high-symmetric structures without CDW and the optimized $2 \times 2 \times 2$ CDW structures are also used to collect 9 more configurations. These diverse sets of 145 configurations are found to be enough to reproduce energy landscape of low-energy stacking structures fruitfully.

The coefficients of basis functions are fitted to the calculated atomic forces. We first perform the linear regression procedure to minimize the following loss function,

$$\sum_i |\mathbf{A}_i \mathbf{C} - \mathbf{F}_i|^2 \tag{3}$$

where $\mathbf{C}$ is the $m$-dimensional coefficient vector of basis functions of $c_{\alpha_1 \alpha'_1 \cdots \alpha_n \alpha'_n}$ in Eq. (2), $\mathbf{F}_i$ is the $3N$-dimensional vector of reference atomic forces in the $i^{\text{th}}$ configuration including $N$ atoms, and $\mathbf{A}_i$ is the $3N \times m$ matrix whose $j^{\text{th}}$ column is atomic forces in the configuration $i$ calculated from the interatomic potential by fixing the coefficient of $j^{\text{th}}$ basis function as $c_j = 1$ and by forcing all the other coefficients to be zeros. Here, $m = m_2 + m_3 + m_4$ where $m_k (k = 2, 3, 4)$ is the number of $k^{\text{th}}$-order basis functions.

For the coefficients of 4th-order basis function to be nonnegative, the coefficient vector of $\mathbf{C}$ obtained from Eq. (3) is again optimized using a nonlinear conjugate gradient method for the modified loss function, $\sum_i |\mathbf{A}_i \mathbf{D} \mathbf{C} - \mathbf{F}_i|^2$ where $\mathbf{D}$ is a $m \times m$ diagonal matrix with conditions of $D_{ii} = -1$ only when $i > m_2 + m_3$ and $c_i < 0$ and otherwise, $D_{ii} = 1$. This enforces nonnegative condition for the coefficients of 4th-order basis function.

## Molecular dynamics simulation

Using the obtained interatomic potential, our molecular dynamics (MD) simulations are performed with the velocity Verlet algorithm[49] for the integration of Newton's equation of motions and simple velocity rescalings are applied for every time steps of 5 fs to incorporate the temperature effect. We have used $60 \times 60 \times 12$ supercell for calculations in the Fig. 2 and $12 \times 12 \times 12$ supercell for Fig. 3. All thermodynamic ensembles are collected after 100ps of thermalization steps. The density-density correlation function is given by $S_{\rho\rho}(\mathbf{k}, \omega; T) = |\rho(\mathbf{k}, \omega; T)|^2$ and $\rho(\mathbf{k}, \omega; T)$ is defined as

$$\rho(\mathbf{k}, \omega; T) = \int_{-\infty}^{\infty} \rho(\mathbf{k}, t; T) e^{-i\omega t} dt,$$

where $\rho(\mathbf{k}, t; T) = \frac{1}{\sqrt{N}} \sum_{l,\kappa} e^{-i\mathbf{k} \cdot \mathbf{R}_{l,\kappa}(t; T)}$ and $\mathbf{R}_{l,\kappa}(t; T)$ is the position vector of basis atom $\kappa$ in a unitcell $\mathbf{R}_l$ at the time $t$ and temperature $T$.

## First-principles calculations

Ab initio calculations based on density functional theory (DFT) are performed using the Vienna ab initio simulation package (VASP)[50] using a plane wave basis set with a kinetic energy cutoff of 300 eV. The generalized gradient approximation with Perdew-Burke-Ernzerhof scheme[51] and dispersion correction using DFT-D3[52] are adopted to approximate exchange-correlation functional and the projector augmented wave method[53] is used for the ionic potentials. For structural optimizations and first-principles MD, $12 \times 12 \times 12$ and $3 \times 3 \times 2$ k-points are sampled in the Brillouin zone of CsV$_3$Sb$_5$, respectively and the internal atomic coordinates of CDW states were optimized until the Hellmann-Feynman forces exerting on each atom becomes less than 0.01 eV/Å.

## Thermal average of potential energy

For anharmonic systems without light elements, ab initio MD simulation is a quite accurate method to investigate its thermal properties. Nevertheless, it is very time-consuming so that less demanding computational methods have been developed. The lists of those methods can be found in the recent literatures[47,54]. The main idea of those methods is that the original system can be approximated by the harmonic systems that minimizes the free energy of the system, and equilibrium structures and effective phonon frequencies (or effective potential) can be obtained from them. Our interest is, however, the dynamics between metastable configurations (i.e. CDW phase domains) so that the effective potential should be obtained by thermally averaging fast motions of atoms not only in the ground state configurations but also in the metastable ones.

To obtain such an average, we first assume that $N$-body density correlation function of $\tilde{\rho}$ does not vary when a ground state thermal ensemble experience a static spatial displacement, i.e., a rigid density approximation. Within this approximation, the static displacement corresponds to the metastable configuration and the thermal average can be performed using the exact $\tilde{\rho}$. Our second approximation is to replace the $\tilde{\rho}$ with $\tilde{\rho}_H$ of an approximate harmonic system, which can be written as a closed form of normal modes[54].

Procedures to obtain the effective potential from the zero-temperature interatomic potential $V$ can be formulated as follows. For periodic systems, $V$ can be expanded with Taylor series of atomic displacements as

$$V = \sum_{n=0}^{\infty} V^{(n)} \equiv \sum_{n=0}^{\infty} \frac{1}{n!} \sum_{\{\alpha\}} \phi_{\alpha_1 \cdots \alpha_n} \prod_{i=1}^{n} u_{\alpha_i} \tag{4}$$

where $\alpha_i$ is a condensed index for Cartesian components, basis atom in a unitcell and the Bravais lattice point. Here, $\phi_{\alpha_1 \cdots \alpha_n}$ is a $n^{\text{th}}$-order force constant and $u_{\alpha_i}$ is a Cartesian component of a displacement vector from a reference position chosen to be a local minimum or saddle point of $V$. The brace indicates that the summation runs for all indices of $\alpha_i$ in the summands.

At finite $T$, the $n^{\text{th}}$-order thermally averaged potential of $V_T^{(n)}$ can be written as

$$\overline{V}_T^{(n)} \equiv \langle V^{(n)} \rangle_T = \frac{1}{n!} \sum_{\{\alpha\}} \phi_{\alpha_1 \cdots \alpha_n} \langle \prod_{i=1}^{n} u_{\alpha_i} \rangle_T \tag{5}$$

where the bracket denotes the ensemble average with $\tilde{\rho}$. Without varying $\tilde{\rho}$, the average potential energy for adding a static displacement of $\mathbf{U} = (U_{\alpha_i}, \cdots, U_{\alpha_n})$ to the thermal ensemble of $u_{\alpha_i}$ can be written as

$$\overline{V}_T^{(n)}(\mathbf{U}) \equiv \frac{1}{n!} \sum_{\{\alpha\}} \phi_{\alpha_1 \cdots \alpha_n} \langle \prod_{i=1}^{n} (U_{\alpha_i} + u_{\alpha_i}) \rangle_T, \tag{6}$$

where $k^{\text{th}}$-order effective force constant is defined as $\phi_{\alpha_1 \cdots \alpha_k} = \frac{\partial^k \overline{V}_T(\mathbf{U})}{\partial U_{\alpha_1} \cdots \partial U_{\alpha_k}}$. When we evaluate $\langle \prod_{i=1}^{n} (U_{\alpha_i} + u_{\alpha_i}) \rangle_T$ in Eq. (6), we replace $\tilde{\rho}$ with $\tilde{\rho}_H$ for which the approximate harmonic system is chosen to have the similar density correlations with the original anharmonic system. Then, using a thermally averaged potential in Eq. (6), we compute a set of harmonic frequencies from which we reconstruct a new $\tilde{\rho}_H$. So, if a self-consistency for $\tilde{\rho}_H$ is fulfilled, we can obtain the desired thermal averaged potential of a rigidly shifted thermal ensemble. We use this method to compute the temperature-dependent interaction parameters of $J_\parallel(T)$ and $J_\perp(T)$ as discussed in Sec. 6 of SI.

## Data availability

All data are available in the main text and the Supplementary Materials.

## Code availability

The code used for molecular dynamics simulation is available upon request.

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

## Acknowledgements
C.P. was supported by the new generation research program (CG079701) at Korea Institute for Advanced Study (KIAS). Y-W.S. was supported by the National Research Foundation of Korea (NRF) (Grant No. 2017R1A5A1014862, SRC program: vdWMRC center) and KIAS individual Grant No. (CG031509). Computations were supported by Center for Advanced Computation of KIAS.

## Author contributions
Y-W.S. conceived the project. C.P. devised numerical methods and performed calculations. C.P. and Y-W.S. analyzed results and wrote the manuscript.

## Competing interests
The authors declare no competing interests.
