## [Peer Review File · Nature Communications]

REVIEWER COMMENTS

Reviewer #1 (Remarks to the Author):

Authors study AV₃Sb₅ characterized by coexistence of CDW and superconducting phase.

Based on the molecular dynamics simulation, the authors suggest that the phase transition to charge ordered phase is a condensation process of incoherently preformed CDWs (in separate kagome layer).

Unfortunately, I have problem with experimental evidence (point A), relation to the direct phonon calculations at finite temperature (point B), and details of the method (point C). IMO presented results can be consequence of presented technique, i.e., can be artificial.

I cannot suggest this paper for publication in the Nature Group journals

A. There is no experimental evidence to 'realization' any structural transition above $T > T_c = 90\text{K}$ for CsV₃Sb₅. Authors give some references in summary, but what is surprising avoid papers containing the Raman scattering: Phys. Rev. Materials 5, L111801 (2021), Phys. Rev. B 104, 165110 (2021), Phys. Rev. Research 4, 023215 (2022), Nat. Commun. 13, 3461 (2022). Discuss there Raman scattering spectra does not exhibit any "additional" peaks above T_c (absence structure modification above T_c). Nevertheless clearly show structural transition at T_c (additional peaks in Raman spectra below T_c).

Similarly, papers discuss NMR suggest modification of intensity for V NMR below T_c (Chinese Phys. Lett. 38 077402 (2021) and npj Quantum Mater. 7, 30 (2022)).

B. Authors avoid in references two papers discuss exact study of the CsV₃Sb₅ dynamical properties, i.e., Phys. Rev. Materials 6, 015001 (2022) and Phys. Rev. B 105, 235134 (2022). In the first one the author analyzed the possible symmetries of CsV₃Sb₅ induced by the soft modes, while in second paper authors presence exact temperature study of the phonon dispersion curves. In the case of second paper, there is strong disagreement between presented here and published there results, i.e., above T_c there is no soft mode. In presented manuscript, authors present phonon spectra with mode softening above T_c . Here differences should be explained.

C. Discuss technique to fitting the interatomic potential is very interesting, but I have some objection. Molecular dynamics simulations were perform at 100 K and 200 K (Section Method) -- however, there is no reason to assumption that the same "fitted" potential is correct the "any" temperature. This can be source of difference with direct calculations (see point B). Also random displacement do not guarantee correct solution (atoms displacement in practice should depends by temperature).

Additionally, what is argument to used the effective interatomic potential, when direct calculations for specific temperature can be perform (with probably smaller computational cost than MD simulation)?

Reviewer #2 (Remarks to the Author):

In this manuscript, the authors perform a combination of density functional theory and molecular dynamics simulations to elucidate the nature of the charge density wave behavior in the AV₃Sb₅ Kagome metal system, with special focus on the CsV₃Sb₅ system. Using a new molecular dynamics method, they find that charge modulation sets in far above the experimentally observed phase transition, which would have valuable implications towards the nature of the CDW. They find that inter-layer charge orderings condense first, with a fluctuating phase relation between neighboring layers that slows with decreasing temperature and eventually stops at T_{CDW} . They find that these fluctuations are larger for larger A sites, and, using the framework of a 4-state Potts model, conclude that this is due to the larger A sites giving rise to larger interlayer spacing. They summarize how these results help explain x-ray diffraction and thermal expansion results at temperatures above T_{CDW} .

Overall, I think a molecular dynamics study of the AV₃Sb₅ Kagome metal system would be an appropriate and timely contribution to the field, and the authors seem capable of performing one. Whether the interlayer periodicity of the CDW state is two, four, or more layers is something I'd be particularly interested in, and would be appropriate for a molecular dynamics simulation since trying to answer that question with density functional theory is typically too computationally intense. However, there are significant parts of this manuscript, both in terms of methodological choices and conclusions drawn, which would need to be clarified or changed before I can recommend it for publication.

1. The behavior of these systems is highly dependent on structural details and convergence. These In the methods section, the authors state that they use a 12x12x12 k-point mesh for structural relaxations of CsV₃Sb₅. This seems to me an odd choice, as the c axis is nearly twice the length of the in-plane lattice parameters, meaning the Brillouin zone should be about half as wide in that direction. Is there a reason why the authors use a much higher k-point sampling density along this direction than the in-plane direction, where most of the displacement associated with the CDW seems to be? The kinetic energy cutoff of 300 eV, especially with the D3 correction, seems very low to me. The force convergence tolerance of 0.01 eV/A, seems very high, especially if the results were

used to fit interatomic potentials for use in molecular dynamics. Were these parameters systematically converged? Can the authors point to other DFT studies which used similar parameters?

2. It isn't clear from the manuscript what size molecular dynamics supercells were used for each part of the study, but this is a very critical parameter. The results of Figure 2 are important to their conclusions, and investigating the L point phonons must involve an MD supercell at least two unit cells deep along the z direction to be meaningful. Based on further conclusions in the manuscript about the importance of coupling between next-nearest layers, it seems like it should probably be at least four. Figure S2 and S6 are the only diagrams I can find that show the MD cell, but they just show a two-dimensional cell. The authors should make the depth of the MD simulation clear. It should also be large enough to accommodate the out-of-plane wavevectors they report on.

3. With respect to the conclusions drawn from Figure 2, I'm confused as to how the authors can see a soft phonon mode at the L point that approaches zero frequency, yet not predict a structural phase change. Their explanation is that "a phase of its order parameter cannot be fixed among the four CDW states irrespective of the fully developed amplitude of CDW order". I'm not sure what that means - could the authors explain this in a different way?

4. The authors highlight the importance of interlayer coupling between next-nearest layers, pointing out that "the tiny magnitude of J_2 is essential in controlling the phase fluctuation with fixed CDW amplitudes across the kagome layers". But they also state that "the accuracy of J_2 from our ab-initio calculations of CsV3Sb5 seems to be marginal", and set it to a constant value. This parameter is clearly important in distinguishing the inter-plane periodicity of the CDW. How can we trust resulting structural predictions with respect to the nature of the CDW phase? And with respect to evaluating the critical temperature, how was the "maximum uncertainty" of 7 K established? Why is this number so small if J_2 is also 'essential'?

5. There has been much recent work linking the CDW transition to time-reversal symmetry breaking phenomena, including loop currents and 'imaginary' charge density waves. It seems as if many of the additional observed property 'kinks' and other signatures of phase transition behavior above and below the accepted T_{CDW} at 94 K have been linked to these fundamentally electronic dynamics (for example, see Christensen, Morten H., et al. "Loop currents in AV3Sb5 kagome metals: Multipolar and toroidal magnetic orders." *Physical Review B* 106.14 (2022): 144504). Since your molecular dynamics study does not explicitly take electronic degrees of freedom into consideration, this kind of correlated behavior seems like something your model cannot capture. Can you make convincing arguments that the behavior you ascribe to atomic fluctuations in the 'Experimental verification' section isn't better explained by these kinds of effects? Is your model still sensible below the T_{CDW} if these kinds of electronic dynamics are present?

Reviewer #3 (Remarks to the Author):

This manuscript by Park and Son studies a kagome metal family AV₃Sb₅ based on first principles computation method. Among many fascinating and intercorrelated properties, authors focus on recent observations of lattice vibration and CDW behavior. The main finding is about the details of CDW formation. Their molecular dynamics simulations indicate the phase fluctuations across the 2D layers and the CDW sets in when it becomes coherent. The known CDW temperature is lower than that of preformed inplane CDW. This interesting new finding is supported by consistent with some recent experiments.

It is a new discovery that provides an useful insight and well deserves much attention from both experimental and theoretical side. Computation methods, procedures and the interpretations are reasonable. To be published in Nature Communications, however, there are some points of improvements. My main concerns are as below.

Data and analysis are useful for understanding one of the main instabilities CDW. But it is not clear how it is related to the other presumably more subtle, difficult and controversial issues in this system. Superconductivity, nematicity and time reversal symmetry breaking phases? Given that all these are discussed in the light of 2d Hubbard models one important step needs to be made to achieve more important progress. Although it is certainly true that interplanar couplings have been observed and the previous 2d theories not fully gratifying, it seems a still vastly open question here and the relevant discussion appears necessary.

The second point is about the technique. Authors used a new MD simulation approach and it provides them the key results. I notice the second not adjacent interlayer interactions are important. The method is well tested and proved to be accurate enough for this very small interaction? I ask about both D3 functional and the elaborate fitting process etc.

Reviewer #4 (Remarks to the Author):

The CDW origin of CVS-family-kagome materials has been debated in recent years. One mechanism is the electron-electron interaction while the competitive mechanism is the lattice instability. Several recent experiments indicate a high possibility of the lattice origin [e.g., Ref.22 - Nature Communications 13, 3461 (2022), arXiv:2301.10299]. However, the lattice instability is seemingly contradicting the absence of soft phonons in some experiments, for example, using hard x-ray scattering (Ref. 8). The CDW origin remained a puzzling unsolved question. In this work, Park and Son studied the CDW transitions with the molecule dynamics simulation, a method relying on interatomic forces and focusing on the lattice dynamics. They succeeded to reproduce the temperature-dependent lattice phase transition. More important, they revealed the phonon softening in the simulations at $k_z = \frac{1}{2}$ and $\frac{1}{4}$, but in at $k_z=0$ plane as the x-ray /neutron experiment measured, explaining why the soft phonon was not detected before. The most interesting result is the competing interlayer CDW order. 2×2 CDW first exists in the plane at $T_{\text{CDW}} < T < T^*$ and later condenses slowly in the c-direction with competition at T_{CDW} . This rationalizes why experiments observed different c-axis ordering ($2 \times 2 \times 2$, $2 \times 2 \times 4$) which is related to different experimental conditions. Although this is somehow expected given the weak interlayer CDW energy, it is more convincing to observe this effect by a dynamical simulation compared to static energetic argument. Therefore, I believe this work provides an insightful input to understand the debated CDW mechanism in kagome metals in a distinct angle (simulation of dynamics) from most theoretical studies in this field. I would strong support to publish this work in a minor revised form.

In the method, I understand the MD parameters were derived from accurate DFT calculations. It is close to the accuracy of fully ab initio. But I do not get how they moved to the 4-state Potts model to analyze T_{CDW} . I am surprised about the accuracy of estimated T_{CDW} , for which ab initio calculations usually perform bad. Although they claimed to derived parameters self-consistently, it is not transparent how ab initio this method is. Which parameters can be generated from MD simulations, which are self-consistently derived, and which are actually empirical? It will be helpful for general readers to understand the reliability of these calculations by explaining more details.

This work focused on the lattice dynamics. A recent time-resolved ARES (arXiv:2301.10299) extracted crucial electronic dynamics by observing the CDW gap. I am curious whether the Authors

can analyze their data and provide helpful insights (like the lattice and related electron relaxation time) to understand this experiment.

Reply to Reviewers – NCOMMS-23-12594-T

Reviewer 1

Authors study AV3Sb5 characterized by coexistence of CDW and superconducting phase. Based on the molecular dynamics simulation, the authors suggest that the phase transition to charge ordered phase is a condensation process of incoherently preformed CDWs (in separate kagome layer). Unfortunately, I have problem with experimental evidence (point 1), relation to the direct phonon calculations at finite temperature (point 2), and details of the method (point 3). IMO presented results can be consequence of presented technique, i.e., can be artificial. I cannot suggest this paper for publication in the Nature Group journals.

We thank Reviewer 1 for his/her careful reading of our manuscript. Several criticisms of Reviewer 1 based on important recent publications would be certainly helpful a lot for clarifying issues in our study of CDW in these interesting kagome metals. Below we provide point-to-point answers to all the questions and concerns of Reviewer 1. We believe that our answers here could resolve all criticisms from Reviewer 1.

Q1. There is no experimental evidence to 'realization' any structural transition above $T > T_c=90\text{K}$ for CsV3Sb5. Authors give some references in summary, but what is surprising avoid papers containing the Raman scattering: Phys. Rev. Materials 5, L111801 (2021), Phys. Rev. B 104, 165110 (2021), Phys. Rev. Research 4, 023215 (2022), Nat. Commun. 13, 3461 (2022). Discuss there Raman scattering spectra does not exhibit any "additional" peaks above T_c (absence structure modification above T_c). Nevertheless, clearly show structural transition at T_c (additional peaks in Raman spectra below T_c). Similarly, papers discuss NMR suggest modification of intensity for V NMR below T_c (Chinese Phys. Lett. 38 077402 (2021) and npj Quantum Mater. 7, 30 (2022)).

A1. First of all, we thank Reviewer 1 for highlighting recent important works on Raman and NMR experiments on kagome metals. Among six papers that Reviewer 1 mentioned, we already cited four works of PRM 5, L111801, Nat. Commun. 13, 3461, Chinese Phys. Lett. 38, 077402 and npj Quantum Mater. 7, 30 as Refs. 23, 22, 35, and 36 of our submitted manuscript, respectively. In a revised version, they are cited as Refs 24, 23, 40 and 39, respectively. We add two remaining works of PRB 104, 165110 and PRR 4, 023215 as Refs. 37 and 38 in a revised version.

We understand that the main concern of the reviewer 1 is on experimental evidences of our predictions of preformed CDWs, especially in Raman scattering and NMR measurements. As we have

mentioned in a section entitled “Discussions”, the preformation is not an actual phase transition but is dynamic fluctuation of incoherent local $2 \times 2 \times 2$ CDW distortions in-between T_{CDW} and T^* . As will be discussed below in detail, the Raman scattering at finite frequencies related with our preformed lattice distortions cannot have finite cross sections. So, the concerns regarding on absence of characteristic “additional” Raman signatures can be resolved well.

Within Placzek’s approximation [Refs. R1, R2, R3, and R4 in the end of Answer1], the non-resonant Raman scattering intensity for the Stokes process can be written as $I \propto \sum_{\nu} |\hat{e}_i \cdot \mathcal{R}_{\nu} \cdot \hat{e}_s|^2$, where \mathcal{R}_{ν} is the Raman tensor associated with the ν -th phonon mode of frequency ω_{ν} , \hat{e}_i and \hat{e}_s are the polarization of incident and scattered lights, respectively. \mathcal{R}_{ν} is proportional to the Raman susceptibility α_{ij}^{ν} , which is defined as $\alpha_{ij}^{\nu} = \sum_{lk} (\partial \chi_{ij}) / (\partial r_{lk}) u_{lk}^{\nu}$. Here, u_{lk}^{ν} is ν -th phonon eigenvector of l -th atom along k -th direction, and χ_{ij} is the electric polarizability tensor ($i, j, k = x, y, z$). For preformed orders, the local in-plane lattice distortion fluctuates between the four different in-plane temporal 2×2 CDWs shown in Fig. 1c of our manuscript. This is well captured by our long-time MD simulations as shown in Fig. 3. The associated folded phonon modes with finite frequencies will experience similar fluctuations. If we assume a possible Raman active in-plane mode of u_{lk}^{ν} from folded phonons in one of fluctuating structures, e.g., soft A_{1g} or E_{2g} modes considered in Fig. 4 of Ref. 22 (Ref. 23 of revised version), the Raman susceptibility averaged over the wavelength of incident light $\overline{\alpha_{ij}^{\nu}}$ should be zero because $\sum_k (\partial \chi_{ij}) / (\partial r_{lk}) u_{lk}^{\nu}$ for the l -th atom from the four different fluctuating structures should cancel each other. Therefore, additional Raman peaks at finite frequencies owing to folded phonons from the preformed CDWs at each layer are not expected to survive or to be Raman-active.

Meanwhile, NMR spectroscopy measurement is not fast enough to resolve the phase fluctuation so that we also expect similar cancellation in NMR signals for the preformed orders. Although the NMR measurements on ^{51}V do not show direct evidences for the preformed orders, they present interesting experimental signatures that actually support our finding indirectly. At temperature right below T_{CDW} , the single resonant peak abruptly split into two broad peaks as shown in Fig. R1 (a) and (b) below, which indicates the amplitude of CDW is not gradually increased from zero at T_{CDW} but has considerably large finite values already at T_{CDW} . This is indeed in sharp contrast to a gradual splitting of NMR peaks across T_{CDW} as shown in usual CDW physics [See Fig. R1 (c) and (d)]. We consider such an unusual first-order transition behavior of NMR peak splitting as an indirect signature of the preformation because the dynamical fluctuation of preformed CDWs could average out split NMR peaks out to a single one above T_{CDW} .

As discussed above, the Raman and NMR spectroscopies would not be so suitable to observe the preformed orders but may hint their indirect signatures.

Figure R1. (a) The temperature dependent ^{51}V NMR transition lines for CsV_3Sb_5 from Nie *et al.*, Nature **604**, 59 (2022). (b) Temperature dependence of the Knight shift difference between two split NMR peaks from Luo *et al.*, npj Quant. Mat. **7**, 30 (2022). (c) NMR transition line shape in the temperature range 250 ~ 140 K for $\text{Rb}_{0.30}\text{MnO}_3$ single crystal of which $T_{\text{CDW}} = 184$ K from Fig. 1 of Butaud *et al.*, Phys. Rev. Lett. **55**, 253 (1985). (d) The obtained temperature dependence of the CDW amplitude from NMR splitting. Fig. 4 of Phys. Rev. Lett. **55**, 253 (1985).

According to Reviewer 1's recommendation, we have added two additional reference that were missed in our original submission and add a sentence as followings.

[REVISION 1: page 13, Discussion Section, highlighted]

[OLD] The preformed charge order discussed so far is not measurable as discontinuities in thermodynamics variables such as a specific heat or susceptibility. But there might be at least three experimental signatures already indicating its existence.

[NEW] The preformed charge order discussed so far is not measurable as discontinuities in thermodynamics variables such as a specific heat or susceptibility. Also, we expect that Raman scattering [23, 24, 37, 38] or nuclear magnetic resonance measurements [39, 40] may not be so easy to capture its explicit signals owing to its fluctuating phases and dynamical nature. But there might be at least three experimental signatures already indicating its existence.

[Ref. R1] Brüesch, P. Phonons: Theory and Experiments II, Springer Berlin (1986).

[Ref. R2] Porezag, D., Pederson, M. R. Infrared Intensities and Raman-Scattering Activities within Density-Functional Theory. Phys. Rev. B **54**, 7830–7836 (1996).

[Ref. R3] Umari, P, Pasquarello, A. Infrared and Raman Spectra of Disordered Materials from First-Principles. Diamond Relat. Mater. **14**, 1255 (2005).

[Ref. R4] Choi, S. M, Jhi, S.-H., Son, Y.-W., Anomalous Optical Phonon Splittings in Sliding Bilayer Graphene, ACS Nano **7**, 7151-7156 (2013).

Q2. Authors avoid in references two papers discuss exact study of the CsV₃Sb₅ dynamical properties, i.e., Phys. Rev. Materials 6, 015001 (2022) and Phys. Rev. B 105, 235134 (2022). In the first one the author analyzed the possible symmetries of CsV₃Sb₅ induced by the soft modes, while in second paper authors presence exact temperature study of the phonon dispersion curves. In the case of second paper, there is strong disagreement between presented here and published there results, i.e., above T_c there is no soft mode. In presented manuscript, authors present phonon spectra with mode softening above T_c. Here differences should be explained.

A2. First of all, we thank Reviewer 1 for pointing out recent theoretical works on phonon dispersions of kagome metals. We have added two works as additional references 29 and 47 in our revised version.

For the first paper (Phys. Rev. Materials 6, 015001 (2022)), we find that it is about the phonon decomposition of ground state of CsV₃Sb₅ at T = 0 and that all their results consistent with our results pointing to 2×2×2 CDW order.

For the second paper (Phys. Rev. B 105, 235134 (2022)), as Reviewer 1 pointed out, we also find that, contrary to ours, their computed lattice dynamics do not show preformed orders above T_{CDW}. The authors of the second paper used ALAMODE code (Phys. Rev. B 92, 054301 (2015)) to compute phonon dispersions at finite temperature. The ALAMODE code obtains interatomic force constants for dynamical matrix by linear regression to reference DFT calculations. This is similar with our approach. However, if compared with ours, there are significant differences in methods for incorporating the temperature effect or higher order anharmonic phonon effects at finite temperature. In ALAMODE code, the anharmonic effects, the most critical components at non-zero temperature, are included for *predefined* initial atomic positions. Specifically, the renormalized phonon frequency ($\Omega_{\mathbf{q}}$) from harmonic one ($\omega_{\mathbf{q}}$) at wavevector \mathbf{q} and temperature of T can be written as $\Omega_{\mathbf{q}}^2 = \omega_{\mathbf{q}}^2 + 2\Omega_{\mathbf{q}}I_{\mathbf{q}}$ where the phonon self-energy of $I_{\mathbf{q}}$ has the fourth-order force constant (Φ) only in the code. So, $I_{\mathbf{q}} = \frac{1}{2}\sum_{\mathbf{p}} \hbar A[n(\Omega_{\mathbf{p}}; T)]\Phi(\mathbf{q}; -\mathbf{q}; \mathbf{p}; -\mathbf{p})/(4\Omega_{\mathbf{q}}\Omega_{\mathbf{p}})$ where $A[n(\Omega_{\mathbf{p}}; T)] = 1 + 2n(\Omega_{\mathbf{p}}; T)$ and $n(\Omega_{\mathbf{p}}; T)$ is the occupation number of phonon mode \mathbf{q} at T . From these equations, one can easily check that the finite temperature phonon modes within ALAMODE code are hardened by repulsive quartic interactions with their thermally average amplitudes of $A[n(\Omega_{\mathbf{p}}; T)]$. Physically, this is equivalent to replace atoms into thermally broadened objects with harmonic and quartic mean-field interactions. It has been well known [See Fig. 5 of the original ALAMODE method paper, PRB 92, 054301] that such a mean-field approach typically enhances the inter-atomic interactions, resulting in overestimation of structural stability or phonon frequencies or in underestimation of critical temperature. This tendency

can be found in their results such that KV_3Sb_5 has no structural instability even at 50 K, quite low temperature than experiment one. Contrary to the method in ALAMODE code, our MD simulation naturally includes not only quartic interactions but also several other 3rd and 4th order phonon-phonon interactions so that the proper soften modes can be reliably described. We also note that this important subject has been recently discussed well by Xiao and Marianetti, Phys. Rev. B **107**, 094303 (2023) (See Fig. 1 and Fig. 3 for comparison of phonon spectral densities between MDs and diagrammatic approaches).

Moreover, due to the diagrammatic approaches of ALAMODE code, this method can compute finite temperature phonon dispersion correctly only when the atomic trajectory can be tracked adiabatically. However, this condition will not hold for multiple trajectories with degenerate energies and the fluctuation of preformed orders in kagome metals seem to be the case. For such a case, the proper treatment of anharmonic interactions at finite T is to compute all possible ensemble averages by gathering all trajectories from long-time stable molecular dynamics. As indeed shown in our MD simulated atomic positions above T_{CDW} in Fig. 3c, thermal fluctuations not only broaden atomic density but also cover all four degenerate positions, that is hardly included in the ALAMODE method.

In summary, disagreements between the results of the second paper and ours originate from the different strategies or methods for including temperature induced anharmonic effects. As we emphasize in the manuscript, the soften modes at higher temperature than T_{CDW} and fluctuations between our preformed order can be captured not by simple diagrammatic computational methods but by our long-time ensemble average.

Q3. Discuss technique to fitting the interatomic potential is very interesting, but I have some objection. Molecular dynamics simulations were performed at 100 K and 200 K (Section Method) – however, there is no reason to assumption that the same “fitted” potential is correct the “any” temperature. This can be source of difference with direct calculations (see point B). Also random displacement do not guarantee correct solution (atoms displacement in practice should depends by temperature). Additionally, what is argument to use the effective interatomic potential, when direct calculations for specific temperature can be perform (with probably smaller computational cost than MD simulation)?

A3. As we have mentioned in the Method section, for the reference DFT calculation, we have used variety of configurations including 100 and 200 K *ab initio* MD results (53 configurations), random displacement (< 0.04 Å) from the optimized $2\times 2\times 2$ CDW structure (30 configurations), and linearly interpolated configurations between high-symmetric structure and the optimized $2\times 2\times 2$ CDW structure (9 configurations). The latter two sets of configurations have lower average potential energy (roughly

20~40% of 200K MD results) and play important roles in reproducing CDW formation energy calculated from DFT.

We would like to emphasize that selective *ab initio* MD calculations for two specific temperatures have been done to sample data in configuration spaces. Our sampling dataset is enlarged further by including more interpolated configurations. The specific temperatures we used could be just matter of choice to improve quality of dataset as usually been done by others also. For example, the second paper that Reviewer 1 mentioned in Q2 used 63 configurations from the thermal ensemble at a single temperature of 150 K and then compute phonon dispersions at different temperatures. As already explained above, the source of disagreements between ours and the second paper in Q2 is not sampling issue but different formalisms to handle higher order anharmonic effects in finite temperature.

The reason why we used the effective interatomic potential is as follows. When a system approaches a critical temperature, the fluctuation becomes to be very slow that we need to wait power-law-increasing time to capture those fluctuations. This is clearly impractical. This critical slowdown near transition temperature had been well known as a notorious problem even in Monte Carlo simulations and special treatments have been developed for reliable results [Refs. R5, R6, R7 and R8 in the end of Answer3]. One of the most successful ways to avoid this problem is Landau-Wang density of state sampling method that we have adopted. Because this method works well for discrete variables, we applied a few controlled approximations including thermal average of potential and mapped continuous CDW amplitudes into 4-state Potts model.

[Ref. R5] Swendsen, R. H., and Wang, J.-S., Nonuniversal Critical Dynamics in Monte Carlo Simulations. Phys. Rev. Lett. **58**, 86-88 (1987).

[Ref. R6] Wolff, U., Collective Monte Carlo Updating for Spin Systems. Phys. Rev. Lett. **62**, 361-364 (1989).

[Ref. R7] Berg, B. A. and Neuhaus T., Multicanonical ensemble: A new approach to simulate first-order phase transitions. Phys. Rev. Lett. **68**, 9 (1992).

[Ref. R8] Lee, J., New Monte Carlo algorithm: Entropic sampling. Phys. Rev. Lett. **71**, 211 (1993).

Reviewer 2

In this manuscript, the authors perform a combination of density functional theory and molecular dynamics simulations to elucidate the nature of the charge density wave behavior in the AV₃Sb₅ Kagome metal system, with special focus on the CsV₃Sb₅ system. Using a new molecular dynamics method, they find that charge modulation sets in far above the experimentally observed phase transition,

which would have valuable implications towards the nature of the CDW. They find that inter-layer charge orderings condense first, with a fluctuating phase relation between neighboring layers that slows with decreasing temperature and eventually stops at T_{CDW} . They find that these fluctuations are larger for larger A sites, and, using the framework of a 4-state Potts model, conclude that this is due to the larger A sites giving rise to larger interlayer spacing. They summarize how these results help explain x-ray diffraction and thermal expansion results at temperatures above T_{CDW} .

Overall, I think a molecular dynamics study of the AV₃Sb₅ Kagome metal system would be an appropriate and timely contribution to the field, and the authors seem capable of performing one. Whether the interlayer periodicity of the CDW state is two, four, or more layers is something I'd be particularly interested in, and would be appropriate for a molecular dynamics simulation since trying to answer that question with density functional theory is typically too computationally intense. However, there are significant parts of this manuscript, both in terms of methodological choices and conclusions drawn, which would need to be clarified or changed before I can recommend it for publication.

First of all, we thank Reviewer 2 for his/her careful reading of our manuscript and encouraging comments. Below we provide detailed point-to-point replies to all questions and comments.

Q1. The behavior of these systems is highly dependent on structural details and convergence. These In the methods section, the authors state that they use a 12x12x12 k-point mesh for structural relaxations of CsV₃Sb₅. This seems to me an odd choice, as the c axis is nearly twice the length of the in-plane lattice parameters, meaning the Brillouin zone should be about half as wide in that direction. Is there a reason why the authors use a much higher k-point sampling density along this direction than the in-plane direction, where most of the displacement associated with the CDW seems to be? The kinetic energy cutoff of 300 eV, especially with the D3 correction, seems very low to me. The force convergence tolerance of 0.01 eV/Å, seems very high, especially if the results were used to fit interatomic potentials for use in molecular dynamics. Were these parameters systematically converged? Can the authors point to other DFT studies which used similar parameters?

A1. We thank Reviewer 2 for asking detailed computational methods in our calculations. We indeed check all parameters consistently throughout computational procedures from DFT to MD and to effective potential methods. The k-point sampling of 12×12×12 gives almost the same total energy differences between configurations with 12×12×6 uniform sampling. Our motivation for the denser sampling along c-axis is to resolve the energy difference between 2×2×2 and 2×2×4 CDW orders up to ~0.1 meV/unitcell, almost the accuracy limit of our calculation. As Reviewer 2 pointed out correctly in Q4 below, this is an important energy scale for interlayer ordering.

The kinetic energy cutoff of 300 eV is confirmed to be sufficient for our pseudopotentials since it is quite higher than the maximum ENCUT parameter of 220 eV for our pseudopotentials. We also note that other DFT works for vanadium based kagome metals also use the same energy cut-off (PRL **127**, 046401 and PRL **130**, 266402). Also, the D3 correction does not depend on the energy cutoff because it does not refer the charge density of the system unlike other van der Waals correction schemes such as Tkatchenko-Scheffler approach. We note that the force criterion of 0.01 eV/Å for the reference calculation seems to be conventional choice for the ground state structure. We also confirm that it plays no role in displacement-force reference data from molecular dynamics.

Q2. It isn't clear from the manuscript what size molecular dynamics supercells were used for each part of the study, but this is a very critical parameter. The results of Figure 2 are important to their conclusions, and investigating the L point phonons must involve an MD supercell at least two-unit cells deep along the z direction to be meaningful. Based on further conclusions in the manuscript about the importance of coupling between next-nearest layers, it seems like it should probably be at least four. Figure S2 and S6 are the only diagrams I can find that show the MD cell, but they just show a two-dimensional cell. The authors should make the depth of the MD simulation clear. It should also be large enough to accommodate the out-of-plane wavevectors they report on.

A2. We thank Reviewer 2 for pointing out our missing information. We have used $60 \times 60 \times 12$ supercell for calculations in the Fig. 2 and $12 \times 12 \times 12$ supercell to capture the slow phase fluctuations in the Fig. 3. We also checked that size effects do not affect the finite-temperature phonon calculations as shown in the Extended Data Fig. 2. We will explicitly include the information in the main manuscript.

[REVISION 2: page 17, Molecular dynamics simulation Section]

[OLD] All thermodynamic ensembles are collected after 100 ps of thermalization steps.

[NEW] We have used $60 \times 60 \times 12$ supercell for calculations in the Fig. 2 and $12 \times 12 \times 12$ supercell for Fig. 3. All thermodynamic ensembles are collected after 100 ps of thermalization steps.

Q3. With respect to the conclusions drawn from Figure 2, I'm confused as to how the authors can see a soft phonon mode at the L point that approaches zero frequency, yet not predict a structural phase change. Their explanation is that "a phase of its order parameter cannot be fixed among the four CDW states irrespective of the fully developed amplitude of CDW order". I'm not sure what that means – could the authors explain this in a different way?

A3. We thank Reviewer 2 for giving us a chance to clarify nature of preformed CDWs. As Reviewer 2 pointed out properly, there are signatures of phonon softening at L-point around 140 K ~ 170 K in Fig. 2c. This indeed indicates a structural instability. The phonon simulations for Fig. 2c have been done for a relatively short time scale of 0.5 nanoseconds. Then, as demonstrated in Fig. 3, owing to fluctuations of CDWs with much longer time scale than one used for Fig. 2c, the soften phonon dispersions with structural instability around 160 K are expected to be fuzzy. Here we present whole simulations in more detail. In Fig. R2 below, we present temporal evolutions of CDWs at 140 K that clearly displays the 2×2 structural distortion as well as consistent π -phase shift (i.e., different colors) between the adjacent layers at most snapshot of molecular dynamics. Regardless of their well-defined structural distortions within a pair of adjacent layers, their global phases do not fix but slowly fluctuate with nanosecond timescale or longer as demonstrated by color changes in Fig. R2 (See caption for further detailed explanations). This slow fluctuation for soft modes at L-point could make their signature at finite frequency be fuzzy but should be manifested as strong broad peaks of density-density correlation function around zero frequency as shown in our Extended Data Figure 2(b) and (d).

Figure R2. Snapshots of molecular dynamics simulation of $12\times 12\times 12$ supercell at 140 K for CsV_3Sb_5 . Each square corresponds to a V layer where each colored filled dot indicates the 2×2 in-plane CDW unitcell. Four colors are the four phases in Fig. 1c. Horizontal axis is time in nanosecond (ns). From top to bottom rows, stacked V layers along c-axis are displayed. Only 5 layers are shown. We set 2×2 CDW as an initial configuration. We can clearly see that at most snapshot, same colors persist on each plane and adjacent layers have different set of colors. This indicates that the local in-plane CDW distortions and relative π -phase shift for adjacent layers is well defined but their global phases or ordering fluctuate with ns timescale.

According to Reviewer 2's comment, we changed our sentence more clearly as follows.

[REVISION 3: page 7, Phonon instability Section]

[OLD] However, the structural instability here does not guarantee phase transition accompanying spontaneous symmetry breaking because a phase of its order parameter cannot be fixed among the four CDWs states irrespective of the fully developed amplitude of CDW order.

[NEW] However, the structural instability here does not guarantee phase transition. Even though the local 2×2 CDW orders within each layer and their π -phase shift between adjacent layers can fully develops, the local CDW orders change their phases dynamically and do not spontaneously break any symmetry of the crystal.

Q4. The authors highlight the importance of interlayer coupling between next-nearest layers, pointing out that “the tiny magnitude of J_2 is essential in controlling the phase fluctuation with fixed CDW amplitudes across the kagome layers”. But they also state that “the accuracy of J_2 from our ab-initio calculations of CsV_3Sb_5 seems to be marginal”, and set it to a constant value. This parameter is clearly important in distinguishing the inter-plane periodicity of the CDW. How can we trust resulting structural predictions with respect to the nature of the CDW phase? And with respect to evaluating the critical temperature, how was the “maximum uncertainty” of 7 K established? Why is this number so small if J_2 is also ‘essential’?

A4. We thank Reviewer 2 for pointing our important aspects in our simulations. We discussed an important role of J_{12} in energetics and in determining the detailed periodicity of CDW orders. However, it turns out that J_{12} does not affect the transition temperature for CDWs seriously. Here we explain its roles further and will revise our manuscript more clearly according to Reviewer 2’s question.

As Reviewer 2 noted, in our 4 states Potts model, J_{12} plays an important role in determining the periodicity of out-of-plane CDW orders, e.g., the $2\times 2\times 2$ vs $2\times 2\times 4$ structures. As we already wrote in a submitted manuscript, within our DFT calculation, we could not precisely determine their values due to tiny energy difference between $2\times 2\times 2$ and $2\times 2\times 4$ structures. The tiny J_{12} is essential in the sense that its non-zero value will determine the CDW periodicity along out-of-plane direction. But it turns out that the precise value of J_{12} is not so critical in determining T_{CDW} . This can be easily checked by setting J_{12} to be quite small, i.e., $|J_{12}| \ll |J_1|$. Owing to the large positive value of J_1 (the nearest neighbor interlayer interaction), the adjacent layer should have clear π -phase shift regardless of small J_{12} value. When we set $J_{12} = 0$, there are still well-defined in-plane CDW orders as well as π -phase shift between a pair of the nearest neighboring layers. With this, we can compute a transition temperature for two-dimensional CDW of a pair of adjacent layers within our 4 states Potts model. From this, we find that the maximum variation of T_{CDW} is about 7 K. We suspect that the small energy scale of J_{12} may be a source of stacking faults in the out-of-plane CDW order and debates on $2\times 2\times 2$ vs $2\times 2\times 4$ CDW structure.

[REVISION 4: page 12]

[OLD] We also note that the accuracy of $J_{\perp 2}$ from our ab initio calculation of CsV₃Sb₅ seems to be marginal. So, we set it a parameter as $J_{\perp 2} = -0.5 J_{\perp 1}$ and confirmed that its variations only cause a maximum uncertainty of ~ 7 K.

[NEW] We also note that the accuracy of $J_{\perp 2}$ from our ab initio calculation of CsV₃Sb₅ seems to be marginal. While the absolute magnitude of $J_{\perp 2}$ determines the ground state stacking structure, it turns out to play a minor role in determining T_{CDW} . So, we set it as a parameter of $J_{\perp 2} = -0.5 J_{\perp 1}$ and checked that T_{CDW} is lowered by at most 7 K when we decrease $J_{\perp 2}$ to be 0.

Q5. There has been much recent work linking the CDW transition to time-reversal symmetry breaking phenomena, including loop currents and ‘imaginary’ charge density waves. It seems as if many of the additional observed property ‘kinks’ and other signatures of phase transition behavior above and below the accepted T_{CDW} at 94 K have been linked to these fundamentally electronic dynamics (for example, see Christensen, Morten H., et al. "Loop currents in AV₃Sb₅ kagome metals: Multipolar and toroidal magnetic orders." *Physical Review B* 106.14 (2022): 144504). Since your molecular dynamics study does not explicitly take electronic degrees of freedom into consideration, this kind of correlated behavior seems like something your model cannot capture. Can you make convincing arguments that the behavior you ascribe to atomic fluctuations in the 'Experimental verification' section isn't better explained by these kinds of effects? Is your model still sensible below the T_{CDW} if these kinds of electronic dynamics are present?

A5. As Reviewer 2 correctly pointed out, our current formalism cannot directly capture loop currents or multipolar magnetic orders originated from electronic or magnetic degree of freedom. If such exceptional mechanisms alter structural orders, we may capture their indirect signature. Our first-principles calculations with usual exchange-correlation functionals do not show any state related with broken time-reversal-symmetry or loop current states. Within our first-principles mean-field level, one important reason for the absence of spinful states is the considerable out-of-plane band dispersion across the Fermi energy, e.g., the bandwidth of V *d*-orbital (Sb *p*-orbital) along LM (Γ A) is ~ 0.5 (1.0) eV [Figure 1 of *Phys. Rev. B* 105, 235145 (2022)]. These large bandwidths along c-axis hardly realize any spin-split states within DFT level of theory.

In experiments, there are several papers reporting disparate temperatures at which time-reversal-symmetry broken states is observed: $\sim T_{\text{CDW}}$ by the Kerr rotation (*Nature Physics* **18**, 1470–1475 (2022)), ~ 50 K in muon spectroscopy (*Phys. Rev. Research* **4**, 033145 (2022)), and ~ 30 –40 K in NMR (*Nature* **604**, 59–64 (2022)) or muon spectroscopy (*Phys. Rev. Research* **4**, 023244 (2022)). So,

for the moment, we cannot compare our mechanism for preformation above T_{CDW} with other magnetic origins because of the wide range of temperatures depending on experimental tools.

We also thank Reviewer 2 for addressing possible extension of our study into lower temperature regime below T_{CDW} and above superconducting critical temperature (T_{SC}). Since our model is capable of computing continuous structural evolution below T_{CDW} , we could in principle calculate structural alternation in low temperature if exotic electronic and magnetic orders couple to bond orders. But even in such a case, we expect that the variation of bond orders might be subtle. We are still investigating whether our interatomic potential model can capture those subtle phase transition or not by combining our structure information at low temperature and DFT calculations.

In summary, our study based on DFT results indicates the most contributing degree of freedom for CDW and the observed anisotropy seems to be phonon (structural) and the electron or magnetic degrees of freedom seems to be minor. We think that it would be better to discuss the emergence of such exotic phases at low-temperature in our future work focusing the temperature range between T_{SC} and T_{CDW} .

Reviewer #3

This manuscript by Park and Son studies a kagome metal family AV_3Sb_5 based on first principles computation method. Among many fascinating and intercorrelated properties, authors focus on recent observations of lattice vibration and CDW behavior. The main finding is about the details of CDW formation. Their molecular dynamics simulations indicate the phase fluctuations across the 2D layers and the CDW sets in when it becomes coherent. The known CDW temperature is lower than that of preformed in-plane CDW. This interesting new finding is supported by consistent with some recent experiments.

It is a new discovery that provides a useful insight and well deserves much attention from both experimental and theoretical side. Computation methods, procedures and the interpretations are reasonable. To be published in Nature Communications, however, there are some points of improvements. My main concerns are as below.

First of all, we thank Reviewer 3 for his/her careful reading of our manuscript and encouraging comments. Below we provide detailed point-to-point replies to all questions and comments.

Q1. Data and analysis are useful for understanding one of the main instabilities CDW. But it is not clear how it is related to the other presumably more subtle, difficult and controversial issues in this system. Superconductivity, nematicity and time reversal symmetry breaking phases? Given that all these are

discussed in the light of 2d Hubbard models one important step needs to be made to achieve more important progress. Although it is certainly true that interplanar couplings have been observed and the previous 2d theories not fully gratifying, it seems a still vastly open question here and the relevant discussion appears necessary.

A1. We thank Reviewer 3 for pointing out intriguing aspects in physics of kagome metals and for asking their relationship with our results. We found that the Question 1 (Q1) of Reviewer 3 partly overlaps with Q5 of Reviewer 2. So, first, we restate our Answer 5 (A5) to Q5 of Reviewer 2 here.

As Reviewer correctly pointed out, our current formalism cannot directly capture loop currents or multipolar magnetic orders originated from electronic or magnetic degree of freedom. However, if such exceptional mechanisms alter structural orders, we may capture their indirect signature. Our first-principles calculations with usual exchange-correlation functionals do not show any state related with broken time-reversal-symmetry or loop current states. Within our first-principles mean-field level, one important reason for the absence of spinful states is the considerable out-of-plane band dispersion across the Fermi energy, e.g., the bandwidth of V *d*-orbital (Sb *p*-orbital) along LM (Γ A) is ~ 0.5 (1.0) eV [Figure 1 of Phys. Rev. B 105, 235145 (2022)]. These large bandwidths along c-axis hardly realize any spin-split states within DFT level theory.

In experiments, there are several papers reporting disparate temperatures at which time-reversal-symmetry broken states is observed: near T_{CDW} by the Kerr rotation (Nature Physics **18**, 1470–1475 (2022)), ~ 50 K in muon spectroscopy (Phys. Rev. Research **4**, 033145 (2022)), and ~ 30 -40K in NMR (Nature **604**, 59–64 (2022)) or muon spectroscopy (Phys. Rev. Research **4**, 023244 (2022)). So, for the moment, we cannot compare our mechanism for preformation above T_{CDW} with other magnetic origins because of the wide range of temperatures depending on experimental tools.

As Reviewer 3 mentioned, all those intriguing phases, e.g., nematicity and time-reversal symmetry breaking phases have been discussed within 2D physics or 2D Hubbard physics. And, also as Reviewer 3 pointed out, our work highlights the importance of interlayer coupling and their fluctuations. Actually, as we already discussed on Reviewer 2's questions, within our first-principles mean-field level, there are no possible spin-polarized states owing to the considerable out-of-plane band dispersion across the Fermi energy, e.g., the bandwidth of V *d*-orbital (Sb *p*-orbital) along LM (Γ A) is ~ 0.5 (1.0) eV [Figure 1 of Phys. Rev. B 105, 235145 (2022)]. These large bandwidths along c-axis hardly realize any spin-split states within DFT level theory. Therefore, our interatomic potential theory could not touch upon the issues on disparate experimental signatures on nematicity and time-reversal symmetry broken phase. However, since our method can compute reliable structural evolutions below T_{CDW} , we could in principle capture structural alternation in low temperature if exotic electronic and

magnetic orders couple to bond orders. But even in such a case, we expect that the variation of bond orders might be subtle. We are still investigating whether our interatomic potential model can capture those subtle phase transition or not.

In summary, our study based on DFT results indicates the most contributing degree of freedom for CDW and the anisotropy seems to be phonon (structural) and the electronic and magnetic degrees of freedom seems to be minor. We think that it would be better to discuss the emergence of such exotic phases at low-temperature in our future work focusing the temperature range between T_{SC} and T_{CDW} .

Q2. The second point is about the technique. Authors used a new MD simulation approach and it provides them the key results. I notice the second not adjacent interlayer interactions are important. The method is well tested and proved to be accurate enough for this very small interaction? I ask about both D3 functional and the elaborate fitting process etc.

A2. We thank Reviewer 3 for pointing out detailed but important points in our methods. We found that the first question of Reviewer 3 overlaps with Q4 of Reviewer 2 and the second one with Q1 of Reviewer 2. So, here, we just re-write our answers of A4 and A1 to Q4 and Q1 of Reviewer 2.

First, as Reviewer 3 properly pointed out, we have thoroughly tested our computational procedures and parameters to obtain reliable energetics and interatomic potentials. The second not adjacent interlayer interaction (J_{12}) is important in determining the out-of-plane CDWs order or the relative stability between the $2\times 2\times 2$ and $2\times 2\times 4$ structures. But it plays minor role in determining the critical temperature of CDW. For example, even without interlayer interaction J_1 and J_{12} , purely two-dimensional ordering can be established at lower temperature than T_{CDW} . In this sense, the “maximum uncertainty of 7 K” in our manuscript is from the difference of T_{CDW} calculated with or without J_{12} . So, for our interatomic method, T_{CDW} can be much reliably calculated than that for ground state stacking configuration which requires highly accurate J_{12} . We added the following sentence in the manuscript to clarify this point.

“While the absolute magnitude of J_{12} determines the ground state stacking structure, it turns out to play a minor role in determining T_{CDW} . So, we set it as a parameter of $J_{12} = -0.5 J_1$ and checked that T_{CDW} is lowered by at most 7 K when we decrease J_{12} to be 0.”

We also confirm that we have thoroughly tested our fitting procedures from DFT results to higher-order interatomic potentials. Comparisons of total energies between DFT and our interatomic potential method are shown in Extended Data Fig. 3 (Supplementary Information Section 7 of revised

version) and their agreements for three systems are well displayed. We also note that the D3 correction does not depend on the energy cutoff because it does not refer to the charge density of the system unlike other van der Waals correction schemes such as Tkatchenko-Scheffler approach. Our choice of parameters for DFT calculations are also found in other recent works on vanadium based kagome metals [PRL **127**, 046401 and PRL **130**, 266402].

Reviewer #4

The CDW origin of CVS-family-kagome materials has been debated in recent years. One mechanism is the electron-electron interaction while the competitive mechanism is the lattice instability. Several recent experiments indicate a high possibility of the lattice origin [e.g., Ref.22 - Nature Communications 13, 3461 (2022), arXiv:2301.10299]. However, the lattice instability is seemingly contradicting the absence of soft phonons in some experiments, for example, using hard x-ray scattering (Ref. 8). The CDW origin remained a puzzling unsolved question. In this work, Park and Son studied the CDW transitions with the molecule dynamics simulation, a method relying on interatomic forces and focusing on the lattice dynamics. They succeeded to reproduce the temperature-dependent lattice phase transition. More important, they revealed the phonon softening in the simulations at $kz = \frac{1}{2}$ and $\frac{1}{4}$, but in at $kz=0$ plane as the x-ray /neutron experiment measured, explaining why the soft phonon was not detected before. The most interesting result is the competing interlayer CDW order. 2×2 CDW first exists in the plane at $T_{CDW} < T < T^*$ and later condenses slowly in the c-direction with competition at T_{CDW} . This rationalizes why experiments observed different c-axis ordering ($2 \times 2 \times 2$, $2 \times 2 \times 4$) which is related to different experimental conditions. Although this is somehow expected given the weak interlayer CDW energy, it is more convincing to observe this effect by a dynamical simulation compared to static energetic argument. Therefore, I believe this work provides an insightful input to understand the debated CDW mechanism in kagome metals in a distinct angle (simulation of dynamics) from most theoretical studies in this field. I would strong support to publish this work in a minor revised form.

First of all, we thank Reviewer 4 for his/her careful reading of our manuscript and encouraging comments. Below we provide detailed point-to-point replies to all questions and comments.

Q1. In the method, I understand the MD parameters were derived from accurate DFT calculations. It is close to the accuracy of fully ab initio. But I do not get how they moved to the 4-state Potts model to analyze T_{CDW} . I am surprised about the accuracy of estimated T_{CDW} , for which ab initio calculations usually perform bad. Although they claimed to derived parameters self-consistently, it is not transparent how ab initio this method is. Which parameters can be generated from MD simulations,

which are self-consistently derived, and which are actually empirical? It will be helpful for general readers to understand the reliability of these calculations by explaining more details.

A1. We appreciate your comments which are greatly helpful to clearly convey our model to general readers. We break your comments into four parts and the corresponding clarifications will be applied in the manuscripts based on the following responses.

First, our estimated T_{CDW} has maximum error of $\sim 15\%$ compared with experiments. This is achievable for estimating melting temperatures [PRB **100**, 014105 (2019)], ferroelectric phase transition [PRB **52**, 6301 (1995)], superconducting transition temperature [Nature **418**, 758 (2002)], and CDW transition temperature [Nature Commun. **12**, 598 (2021)] using modern *ab initio* based-approaches if including proper anharmonic and temperature effects, and our approach shows similar performance.

Second, with the following step-by-step approximations, we can proceed from full atomic pictures to 4-states Potts model with temperature-dependent exchange interaction as follows. The first approximation is to reduce all atomic degrees of freedom in 2×2 CDW into amplitudes and phases of a three low energy modes M_1 , M_2 and M_3 as shown in Fig. 1b. Because eigenvectors of the three modes can be made real (all atomic vibrations are in-phase), phases can be absorbed into amplitudes as signs and three (continuous, signed) amplitude variable describes the low energy structures. Then, we ignore high energy fluctuation effect and then construct a model for CDWs with three internal degrees of freedom and local interactions between different CDW modes. But, as far as we know, no reliable and efficient Monte Carlo method is available to compute phase transition within this model. So, we further reduced the degree of freedom by considering that the thermal distribution of CDW eigenmode amplitudes [m_1 in y -axis of Fig. 3a] has two well separated distributions below 160 K. In other words, Fig. 3a shows very fast fluctuating red and blue dots around $m_1 = +0.1$ and $m_1 = -0.1$, respectively, indicating thermal vibrational of CDW eigenmodes while it also shows several jumps between red and blue dots with a much longer times scale, indicating phase fluctuations between difference CDW phases shown in Fig. 1c. Then, we exploit the two well-separated time scales of dynamics (thermal vibration and phase fluctuation in Fig. 3a) and further approximate the former into discrete amplitude variables with temperature dependences originated from the thermal average procedure as discussed in a method section entitled “Thermal average of potential energy”. Finally, for the signs of M 's are not independent but energetically favor for their multiplied sign being positive (see Fig. 1c), this results in 4-states Potts model.

Third, as discussed in our manuscript, there are no empirical parameters in our calculation except $J_{\perp 2}$. We find that accuracy of our first-principles method is marginal for this parameter. So, we

set $J_{12}(T) = -0.5J_{\perp}(T)$ and have discussed its impact and roles thoroughly in our manuscript and again in our answers to questions from Reviewer 2 and Reviewer 3 above. All the other parameters for 4-state Potts model at finite temperature can be obtained self-consistently from first-principles results and interatomic potentials as outlined in “Thermal average of potential energy” section in Method part. As shown in the section and also discussed in A2 to Q2 of Reviewer 1, our method is beyond the conventional mean-field approach including the higher-order anharmonic effects at finite temperature and all parameters are self-consistently obtained except one for 4 states Potts model for T_{CDW} .

Finally, there is another self-consistency relation in determining the critical temperature from the 4-states Potts model (This relation has nothing to do with the self-consistency relation used for the above thermal average procedure). A usual 4-states Potts model with fixed exchange parameters can be handled very well using Landau-Wang Monte Carlo method as we discussed in our A3 to Q3 of Reviewer 1. However, to compute T_{CDW} of our 4-states Potts model, we need a self-consistency owing to temperature-dependent exchange parameters. We discussed this self-consistency in a section entitled “Critical temperature of charge orders” in the main manuscript. We think the two self-consistency relations may confuse readers so that we will revise our manuscript as follows.

[REVISION 5: page 10, Critical temperatures of charge orders Section]

[OLD] The four phases of 2×2 CDW in Fig. 1c become 4-states ‘spin’ variables on lattice points of layered triangular lattices as shown in Fig. 4a and the energy of a domain wall between different phases becomes their ‘interaction’ parameters.

[NEW] Due to the two well-separated time scales of dynamics (thermal vibration and phase fluctuation in Fig. 3a), if we average the atomic trajectories of preformed CDW’s over ~ 100 ps, the averaged snapshot will look like one of four 2×2 CDW phase in Fig. 1c, but with temperature-dependent amplitudes. The four phases become 4-states ‘spin’ variables on lattice points of layered triangular lattices as shown in Fig. 4a.

[REVISION 6: page 11, Critical temperatures of charge orders Section]

[OLD] Unlike typical Potts models [27], interaction parameters in Eq. 1 are explicit functions of temperature because amplitudes of CDW vary as the temperature changes. So, T_{CDW} from Eq. 1 should be determined to satisfy self-consistent conditions for $J_{\parallel}(T)$, $J_{\perp}(T)$, $J_{12}(T)$ and $T_{CDW}(J_{\parallel}, J_{\perp}, J_{12})$ simultaneously. The $J_{\parallel}(T)$ and $J_{\perp}(T)$ can be estimated from the potential energy costs to form a CDW phase domain (See detailed procedures in SI. Sec. 4). Because the potential energy is thermally

fluctuating, we averaged it using harmonic and rigid approximations for density correlation functions (See Methods).

[NEW] Unlike typical Potts models [27], interaction parameters in Eq. 1 are explicit functions of temperature because amplitudes of CDW vary as the temperature changes. For $T = 0$, $J_{\parallel}(0)$ and $J_{\perp}(0)$ can be readily calculated from the energy cost forming domain wall and energy differences between different stacking, respectively (See detailed procedures in SI. Sec. 6). For $T > 0$, they are similarly obtained from thermally averaged potential energy of domain structure. Here, the thermal average is approximately calculated using harmonic and rigid approximations for density correlation functions (See Methods). Then, we have temperature-dependent exchange parameters of $J_{\parallel}(T)$, $J_{\perp}(T)$, and $J_{\perp 2}(T)$ in our 4-states Potts model. So, to compute T_{CDW} , we need to solve the model self-consistently because temperature and the exchange parameters depend on each other.

Q2. This work focused on the lattice dynamics. A recent time-resolved ARPES (arXiv:2301.10299) extracted crucial electronic dynamics by observing the CDW gap. I am curious whether the Authors can analyze their data and provide helpful insights (like the lattice and related electron relaxation time) to understand this experiment.

A2. We thank Reviewer 4 for pointing out an important recent experiment from Nuh Gedik's group on CDW of CsV_3Sb_5 (arXiv:2301.10299). First of all, our results support their main conclusion, "structural rather than electronic mechanism of charge density wave." They clearly demonstrated that excited in-gap states at M-point couple to bosonic modes with two characteristic frequencies. The two frequencies measured at M-point has not been observed in previous Raman studies and shows strong dependence on fluence of applied pump pulses. They attribute them as CDW amplitude modes related with star of David (SOD) and inverse star of David (iSOD) shaped lattice modulations, respectively.

If one may assume that the melted CDW states by a pump pulse have some similarities with a gapless or pseudogap states far above T_{CDW} , various low energy soft modes at L- and U-points around preformation temperature shown in our study (Fig. 2c and Extended Data Fig. 1b (Fig. S2(b) for revised version)) could have some relevance with their observations. These soft modes should couple to electronic states at M-point so that they could play a role for decay channels for excited states at M-point. However, excited states by pump pulse actually alter screenings significantly such that one need to be very careful to compute renormalized phonon frequencies due to excited electrons and holes. Therefore, we are not so certain that our results for various high temperature soft modes around T^* have some relevance with the experiments or not. Beside these aspects, from all our simulation results, we do not see any signature from SOD shaped lattice distortion because its energy scale is too high to be accessed by higher order phonon interactions through thermal excitations. This is also consistent with

recent NMR measurements showing no evidence for NMR peak splitting related with SOD pattern, e.g., Chinese Phys. Lett. **38**, 077402 (2021), npj Quantum Mater. **7**, 30 (2022), and Nature **604**, 59 (2022). All in all, we expect that, with careful inclusion of another phonon self-energy correction from electron-hole excitation, our formalism could resolve some puzzles from this important experiment eventually in near future.

REVIEWER COMMENTS

Reviewer #1 (Remarks to the Author):

I would like thank Authors to their reply.

Unfortunately, the Reply does not solve my objection, and I still cannot recommend this paper to publication in Nature Communication.

A. Inconsistency with previous calculation.

Mentioned earlier study (Phys. Rev. B 105, 235134) present results for 150 K and 50 K. Calculations were perform withing ALAMODE software, which in recent version estimate harmonic and high order inter-atomic force constant (IFC), see https://alamode.readthedocs.io/en/latest/almdir/formalism_alm.html .

Maximal order and range of interaction of the IFC is define by several parameters (i.e. NORDER, NBODY, and cutoff, see in <https://alamode.readthedocs.io/en/latest/almdir/inputalm.html>).

IFC are calculated from "several" samples, corresponding to some finite temperature (here we should note that the displacement of the atoms also depends by temperature).

From this, we should treat results from Phys. Rev. B 105, 235134 as a reference results (in this paper we can find information that the authors "included both harmonic and higher-order contributions to phonons").

Reply to the 2nd point did not convince me.

There is no reason, why results for the same temperature should exhibit large "deviation" from this results.

Both calculations contains the 3rd and 4th order phonon-phonon interaction, while only sampling method is not the same.

IMO this is important problem, which should be more carefully explain before acceptance.

In method section authors tell about sampling of the system by MD at 100 and 200 K.

This sampling can be used to calculations directly the phonon dispersion at this temperatures.

Unfortunately, on figure 2 we cannot find results for this temperature -- I strongly suggest this type of calculation as a stress test of presented results.

B. Method

Unfortunately, after replay to 3rd point, I'm more confusing.

Section describing "Construction of interatomic potential" should be improved.

At this moment, paragraph "The reference configurations..." is confusing -- i.e.: 1st sentence is too long and not explaining how configurations were collected.

MD of 4x4 supercell (here is typos probably should be 4x4x4 or 4x4x1) at 100 and 200 K (53 configurations) -- why 53 configurations? or there were initial 53 configurations for MD simulation? similarly: random displacement correspond to the initial atom positions? this should be more carefully described before acceptance.

Reviewer #2 (Remarks to the Author):

I thank the authors for their thoughtful responses to my points, and believe the proposed changes are sufficient for me to recommend publication. I still believe they should take more care in the future to converge their DFT parameters, and not just with respect to total energy, but with respect to forces and structural parameters as well. For example, it will not always be sufficient to choose a cutoff energy based on the ENCUT suggested by the pseudopotential - convergence here often depends on structural and bonding details that the creator of the pseudopotential cannot anticipate. That said, I believe their molecular dynamics experiment is well designed and executed, and will be a valuable contribution to the literature.

Reviewer #3 (Remarks to the Author):

For Q2: I think authors have provided the reasonable reply to my and other reviewers' concerns on their computation method used in this study, and the manuscript revised reasonably well.

For Q1: The reply provides the clear and useful comments about their methodological limitations and relevant reference citations, and it might also be true in regard to the physical interpretation and implication. For a large portion of readership, I think, this issue can be the most important part of discussion as also suggested by other reviewers. But it doesn't seem to be updated in the revised manuscript. Given that authors can make sound and comprehensive discussion about the implication of their findings in the context of or in the relation to the mainline viewpoints, the current work would well be suitable for Nat Comm.

Reviewer #4 (Remarks to the Author):

The Authors properly addressed my questions.

Furthermore, they successfully reproduced the structure phase transition using MD without considering the electronic degree of freedom. From my perspective, it is convincing enough to exclude the electronic origin (like strong correlation) as the main driving force for CDW.

Reply to Reviewers – NCOMMS-23-12594-A

Reviewer #1 (Remarks to the Author):

I would like thank Authors to their reply. Unfortunately, the Replay does not solve my objection, and I still cannot recommend this paper to publication in Nature Communication.

We thank Reviewer 1 for his/her careful reading of our revised manuscript and replies. We believe that our replies could resolve several previous criticisms of Reviewer 1. Below, we provide our point-to-point answers to remaining few questions from Reviewer 1.

A. Inconsistency with previous calculation.

Mentioned earlier study (Phys. Rev. B 105, 235134) present results for 150 K and 50 K. Calculations were perform withing ALAMODE software, which in recent version estimate harmonic and high order inter-atomic force constant (IFC), see https://alamode.readthedocs.io/en/latest/almdir/formalism_alm.html .

Maximal order and range of interaction of the IFC is define by several parameters (i.e. NORDER, NBODY, and cutoff, see in <https://alamode.readthedocs.io/en/latest/almdir/inputalm.html>).

IFC are calculated from "several" samples, corresponding to some finite temperature (here we should note that the displacement of the atoms also depends by temperature). From this, we should treat results from Phys. Rev. B 105, 235134 as a reference results (in this paper we can find information that the authors "included both harmonic and higher-order contributions to phonons").

We thank Reviewer 1 for providing us a chance to explain again the difference between our method and ALAMODE code used in PRB 105, 235134 (or perturbation approaches using a limited number of Feynman diagrams in general). As we explained in our previous answer of A2 to Q2 of Reviewer 1, the cubic and quartic phonon interactions from a few Feynman diagrams are insufficient to describe structural stability or phonon frequency renormalizations at critical temperature. Instead, the molecular dynamics can be used to assess the range of validity for such diagrammatic approaches, not vice versa. Moreover, as we also discussed before, the diagrammatic approaches there cannot handle trajectories with multiple centers (*i.e.*, multi-peak structure in atomic density) because they compute phonon renormalizations for fixed atomic positions. So, these aspects in the earlier study that Reviewer 1 mentioned inevitably result in inaccurate phonon dispersions (or no instability to CDW) for KV_3Sb_5 even at very low temperature of 50 K (far below the transition temperature of its CDW phase). And, they also cannot describe fluctuating phases at all. These contradictions do not originate from simple differences in choosing computational parameters or in sampling strategies but from the fundamental incompetency in perturbative methodologies. We believe that these general aspects have been discussed already in other studies as we wrote in our previous reply. Therefore, we disagree with Reviewer 1's opinion of "treating results from PRB 105, 235134 as a reference result".

Replay to the 2nd point did not convince me. There is no reason, why results for the same temperature should exhibit large "deviation" from this results. Both calculations contains the 3rd and 4th order phonon-phonon interaction, while only sampling method is not the same. IMO this is important problem, which should be more carefully explain before acceptance.

As we discussed in our previous replies of A2 and A3 to Q2 and Q3 from Reviewer 1 as well as our answers above, the large “deviation” is not from the sampling differences but from the methodological ones.

In method section authors tell about sampling of the system by MD at 100 and 200 K. This sampling can be used to calculations directly the phonon dispersion at this temperatures. Unfortunately, on figure 2 we cannot find results for this temperature -- I strongly suggest this type calculation as a stress test of presented results.

As we already discussed in our answer of A3 to Q3 of Reviewer 1, the previous results in PRB 105, 235134 did sample 63 configurations at a single temperature of 150 K. This kind of sampling is quite usual in computing higher-order phonon-phonon interactions in phase spaces at finite temperatures. It could be much better if one can sample more diverse configurations to obtain reliable potentials as we did. However, the samplings at some specific temperatures cannot limit or guarantee accuracy of higher order phonon interactions directly. So, none of these can be a stress test.

According to Reviewer 1’s request, we performed computations on phonon dispersions in whole temperature ranges of interest. Actually, we already present dispersion results at 200 K in our previous Supplementary Information. Below, we put our results for dispersions at 100 and 200 K, respectively, together with Fig. 2. As expected, the dispersions at 200 K indeed show softening of phonons at L point and do not exhibit phonon softening at 100 K. As shown below, the additional dispersions at 100 K and 200K do not represent any special case so that we’d better maintaining Fig. 2 as it is since the dispersions at 200 K are already in our Supplementary Information.

B. Method

Unfortunately, after replay to 3rd point, I'm more confusing. Section describing "Construction of interatomic potential" should be improved. At this moment, paragraph "The reference configurations..." is confusing -- i.e.: 1st sentence is too long and not explainer how configurations were collected.

MD of 4x4 supercell (here is typos probably should be 4x4x4 or 4x4x1) at 100 and 200 K (53 configurations) - why 53 configurations? or there were initial 53 configuration for MD simulation? similarly: random

displacement correspond to the initial atom positions? this should be more carefully described before acceptance.

We thank Reviewer 1 for suggesting an improvement in descriptions of our computational procedures in Method sections. We rewrite the sentence of “The reference configuration...” and put more descriptions as follows in our revised manuscript.

In the third paragraph of page 16:

The reference configuration structure (9 configurations).

→ The reference configurations are collected from multiple sources to ensure reliable sampling of potential energy surfaces. We performed first-principles MD simulations of $4\times 4\times 2$ supercell at 100 and 200 K for 1600 steps with a time step of 10 fs. We then collected 53 configurations from each temperature that were equally spaced in time of 300 fs. Additional 30 configurations are collected from randomly displacing (< 0.04 Angstrom) atomic positions from the optimized $2\times 2\times 2$ CDW structure. Lastly, the linearly interpolated configurations between high-symmetric structures without CDW and the optimized $2\times 2\times 2$ CDW structures are also used to collect 9 more configurations. These diverse sets of 145 configurations are found to be enough to reproduce energy landscape of low-energy stacking structures fruitfully.

Reviewer #2 (Remarks to the Author):

I thank the authors for their thoughtful responses to my points, and believe the proposed changes are sufficient for me to recommend publication. I still believe they should take more care in the future to converge their DFT parameters, and not just with respect to total energy, but with respect to forces and structural parameters as well. For example, it will not always be sufficient to choose a cutoff energy based on the ENCUT suggested by the pseudopotential - convergence here often depends on structural and bonding details that the creator of the pseudopotential cannot anticipate. That said, I believe their molecular dynamics experiment is well designed and executed, and will be a valuable contribution to the literature.

We thank Reviewer 2 for his/her recommendation of our manuscript. We acknowledge Reviewer 2’s opinion on a stringent convergence test for forces and structural parameters beyond typical usages. In future works, we will report further scrutinized tests for all DFT parameters of vanadium kagome compounds beyond the conventional values.

Reviewer #3 (Remarks to the Author):

For Q2: I think authors have provided the reasonable reply to my and other reviewers’ concerns on their computation method used in this study, and the manuscript revised reasonably well.

For Q1: The reply provides the clear and useful comments about their methodological limitations and relevant reference citations, and it might also be true in regard to the physical interpretation and implication. For a large portion of readership, I think, this issue can be the most important part of discussion as also suggested by other reviewers. But it doesn’t seem to be updated in the revised manuscript. Given that authors can make sound and

comprehensive discussion about the implication of their findings in the context of or in the relation to the mainline viewpoints, the current work would well be suitable for Nat Comm.

We thank Reviewer 3 for his/her recommendation of our manuscript. According to Reviewer 3's suggestion on Q1, we add a following new paragraph in Discussion section on our methodological limitations and add one new reference of Phys. Rev. B 105, 235145 (2022) that we cited in our previous reply as Ref. 43.

In the third paragraph of page 14:

We note that our current MD methods from first-principles approaches do not show any state related with broken time-reversal-symmetry states. Within our methods, the vanadium *d*-orbitals have considerable out-of-plane band dispersion across the Fermi energy [43] so that spin-split states related with them hardly develop. From these, our MD could not touch upon the various experimental signatures [1-3,11-13] on time-reversal-symmetry broken phases. However, if exotic electronic and magnetic orders can couple to bond orders below T_{CDW} , we expect that our method could reflect the corresponding structural evolutions at lower temperature.

Reviewer #4 (Remarks to the Author):

The Authors properly addressed my questions. Furthermore, they successfully reproduced the structure phase transition using MD without considering the electronic degree of freedom. From my perspective, it is convincing enough to exclude the electronic origin (like strong correlation) as the main driving force for CDW.

We thank Reviewer 4 for his/her recommendation of our manuscript.